# Multi-layered ecological interactions determine growth of clinical antibiotic-resistant strains within human microbiomes

Ricardo Leon-Sampedro [1] ✉, Mathilde Boumasmoud [1], Markus Reichlin [1], Katia R. Pfrunder-Cardozo[1], Nicholas Noll[2], Adrian Egli[3] & Alex R. Hall [1]

The spread of antibiotic-resistant bacteria in the gut depends on their ability to establish within complex microbial communities. However, the role of various ecological factors in modulating this process, particularly in the absence of antibiotic selection, remains poorly understood. We hypothesize that different strains within the same species vary in their ability to colonize due to distinct interactions with resident microbiota. Using human gut-microbiome samples in replicated anaerobic microcosms with and without antibiotics, we test multiple clinically relevant and phylogenetically distinct *Escherichia coli* strains carrying extended-spectrum beta-lactamase (ESBL) or carbapenemase plasmids. While antibiotics influence the growth of incoming resistant strains, some are successful even without antibiotics. Growth outcomes depend on a combination of intrinsic growth capacities in relevant abiotic conditions, competition with resident *E. coli*, and strain-specific shifts in resident community composition. We also detect horizontal transfer of resistance plasmids in some conditions, but transconjugants remain rare across treatments. Here, we show that the success of antibiotic-resistant bacteria depends on strain-specific ecological interactions, helping to explain the spread and persistence of resistance in human microbiomes.

Combatting antibiotic resistance is a pressing medical challenge worldwide[1,2]. Resistance can spread when bacteria are exposed to antibiotics, such as during treatment of infections. However, some resistant strains appear to be capable of disseminating even in the absence of direct exposure to selecting antibiotics, such as during asymptomatic carriage of *Escherichia coli* strains producing extended-spectrum beta-lactamases (ESBLs) (e.g., sequence type (ST) 131)[3–5] or methicillin-resistant *Staphylococcus aureus* (MRSA) (e.g., MRSA US300)[6–8]. Dissemination of associated resistance genes can occur through vertical transmission and via horizontal gene transfer, often mediated by plasmids. These mobile genetic elements play a central role in the acquisition and spread of resistance determinants[9]. Notably, some plasmid-strain combinations have been linked to increased transmission potential and epidemiological success[10,11]. In addition to

influencing the colonisation success of their bacterial hosts, conjugative plasmids can also spread resistance by transferring into resident bacteria in the gut microbiome[10,12]. This horizontal gene transfer can occur independently of strain colonisation, potentially allowing resistance determinants to persist even when the incoming strain is outcompeted. Carriage of such strains is linked to worse clinical outcomes and risks of later infection and transmission, making it an important obstacle in the context of the overall resistance problem[13,14]. A particular concern is the emergence of resistant clones associated with nosocomial infections in hospital environments, where bacterial transmission between patients is mainly mediated by high-risk clones[10,11]. Although antibiotic exposure is a major driver of resistance selection, resistant bacteria can persist and spread in the absence of antibiotics, supported by low fitness costs, compensatory

[1]Institute of Integrative Biology, Department of Environmental Systems Science, ETH Zurich, Zurich, Switzerland. [2]Karius, Redwood City, California, USA. [3]Institute of Medical Microbiology, University of Zurich, Zurich, Switzerland. ✉ e-mail: rlsampedro85@gmail.com

evolution, or ecological interactions that mitigate the burden of resistance[12,15–17]. Combatting the spread of resistance, therefore, requires that we understand what drives the spread or decline of newly arriving (and potentially colonising) resistant strains in individual human microbiomes, both during treatment and without antibiotics, even in cases when they displace resident *E. coli*[16,17]. Multiple layers of ecological interactions may play a role[18], including bacterial intrinsic population growth in local abiotic conditions[19,20], and interactions with other strains and species[21–24]. However, directly observing such interactions during colonisation by antibiotic-resistant strains in human microbiomes, and separating the role of microbial interactions from other factors that vary among individual hosts[25], is challenging. Consequently, the ecological processes by which different incoming resistant strains interact with resident microbiota, and how these interactions influence their population growth and establishment, are not fully understood.

Identifying which ecological factors determine population growth of different incoming resistant bacteria in individual microbiomes would advance our basic understanding of the ecological drivers of microbial invasions[26,27]. It would also improve our ability to track and manage the spread of resistance: (i) for example, if invasion success is closely linked to intrinsic growth performance in local abiotic conditions[28], this can potentially be measured by pure-culture assays with relevant resistant strains in appropriate conditions. (ii) If competitive interactions with resident strains of the same species play a key role[21,29–31], these might not be captured by pure-culture assays with incoming strains alone. Instead, isolating and profiling resident conspecifics alongside potentially invading resistant strains would be more informative, for example, via strain-level sequencing and phenotyping. (iii) Finally, if invasion success involves extensive interactions with other species[24,32,33], then accounting for the taxonomic composition of the resident microbiota, for example by metagenomic sequencing[34], may be more informative. There is some evidence for each of these drivers, for example, from experiments with mice[21] and assembled communities[24]. Yet replicated experimental tests with clinically relevant resistant strains in microbiome samples from humans are lacking, leaving an important gap in our knowledge of the ecological constraints that limit the dissemination of resistance among individuals.

Here, we use replicated anaerobic microcosms to track the population growth of four clinical strains *E. coli*, with their respective antibiotic-resistance plasmids in gut microbiome samples from healthy humans (Fig. 1A, B and Supplementary Table 1 and 2), aiming to identify the factors and interactions that drive their invasion. We hypothesised that different strains within the same species would exhibit different interactions with resident microbiota, leading to variable invasion outcomes. To test this, we use multiple clinical strains and a controlled design with uninvaded microcosms to assess how growth performance and interactions with resident microbiota varied among incoming strains, potentially revealing mechanisms linked with colonisation success. Each strain belongs to one of the four most abundant phylogroups in humans[35–37]. The four strains represent different sequence types (STs) and carry different clinically relevant resistance plasmids (encoding Extended-Spectrum β-Lactamases -ESBLs- and carbapenemases) (Supplementary Table 2). This approach allows us to overcome some of the key challenges from past work on drivers of invasion in human microbiomes. First, it enables us to make replicated experiments with individual microbiome samples in multiple conditions, isolating the contributions of strain-to-strain and microbiome-to-microbiome variation. We monitor these effects during a two-day experiment, modelling the initial phase of the invasion process[27]. Second, the microbial community in each microcosm is derived from a human gastrointestinal tract, potentially accounting for taxa that are not present in experiments with non-human-associated microbiota or model communities. Third, the use of clinical strains and

plasmids accounts for possible genomic factors that may influence colonisation success, such as pathogenicity islands, prophages or other plasmids, which are often not represented in domesticated or model strains. Including four phylogenetically diverse, clinically relevant strains with distinct genomic and ecological features (Fig. 1 and Supplementary Fig. 1) allows us to detect repeatable patterns across independent microcosms, despite the inherent complexity of human-associated microbial communities. We find variable and strain-specific growth success, with some strains showing positive population growth in human microbiome samples even in the absence of antibiotics. This is positively correlated to the variable growth capacity of the incoming strains across different strain-microbiome combinations in local abiotic conditions, but also to strain-specific interactions with resident microbiota, both within the same species and between different species.

## Results

### Growth success of resistant strains depends on strain and microbiome

We introduced four phylogenetically distinct clinical strains of *E. coli* with their respective beta-lactamase encoding antibiotic-resistant plasmids (Supplementary Table 1 and Fig. 1) into anaerobically incubated microbiome samples (gut microcosms, see Methods) from three healthy human donors, and we did this with and without a low concentration of ampicillin (hereafter referred to as the 'live' microcosm assay). In the presence of ampicillin, all four introduced resistant strains underwent net positive population growth over 48 h (paired $t$ test for each strain; Ec040: $t = 15.1$, df = 2, $p < 0.001$; Ec069: $t = 8.82$, df = 2, $p < 0.001$; Ec131: $t = 3.65$, df = 2, $p = 0.0065$; Ec744: $t = 5.35$, df = 2, $p < 0.001$; Fig. 1C). In these conditions, different resistant strains reached similar population densities, and this was consistent across gut microcosms from different human donors (two-way ANOVA, strain effect: $F(3,24) = 1.835$, $p > 0.05$; donor effect: $F(2,24) = 1.758$, $p > 0.05$; strain × donor interaction: $F(6,24) = 2.275$, $p > 0.05$). By contrast, in the absence of antibiotics (Fig. 1D), net population growth was more variable (strain effect for final abundance: $F(3,24) = 22.438$, $p < 0.001$). For example, Strain Ec040 (sequence type (ST) 40, carrying an ESBL-plasmid- $bla_{CTX-M-1}$) consistently showed net positive growth in gut microcosms from all three human donors, but strain Ec744 (ST744, carrying a carbapenemase-plasmid- $bla_{OXA-48}$) never did (Fig. 1D). The differences among strains also varied among gut microcosms from different people (strain × donor interaction: $F(6,24) = 2.623$, $p < 0.05$). For example, Ec069 (ST69, carrying an ESBL-plasmid- $bla_{CTX-M-14}$) and Ec131 (ST131, carrying a carbapenemase-plasmid- $bla_{KPC2}$) failed to grow in Donor1's gut microcosms, but they were successfully maintained in microcosms from the other donors. Despite this, we found a similar qualitative trend (strain ranking) and average population density of introduced strains across the three donor microcosms (donor effect: $F(2,24) = 2.428$, $p > 0.05$). A separate experiment indicated that variable final abundances of the incoming strains were unlikely to be explained by variable inoculum densities or early-phase abundances (Supplementary Fig. 2). In summary, we found the final abundance of incoming antibiotic-resistant *E. coli* (various clinical strains carrying their natively-associated resistance plasmids) varied among different strains, with some being successful even in the absence of antibiotics.

### Strain-specific population growth in local abiotic conditions

A possible driver of variable ecological success (Fig. 1) is variation of introduced strains' intrinsic population growth capacities in local abiotic conditions. We measured this in a sterilised-microcosm set-up, using autoclaved slurries of the same microbiome samples and the same resistant strains (see "Methods"). This revealed net-positive population growth by all strains in microcosms produced using autoclaved faecal slurries from the same human donors, with resistant strains consistently reaching high final abundances (mean abundance

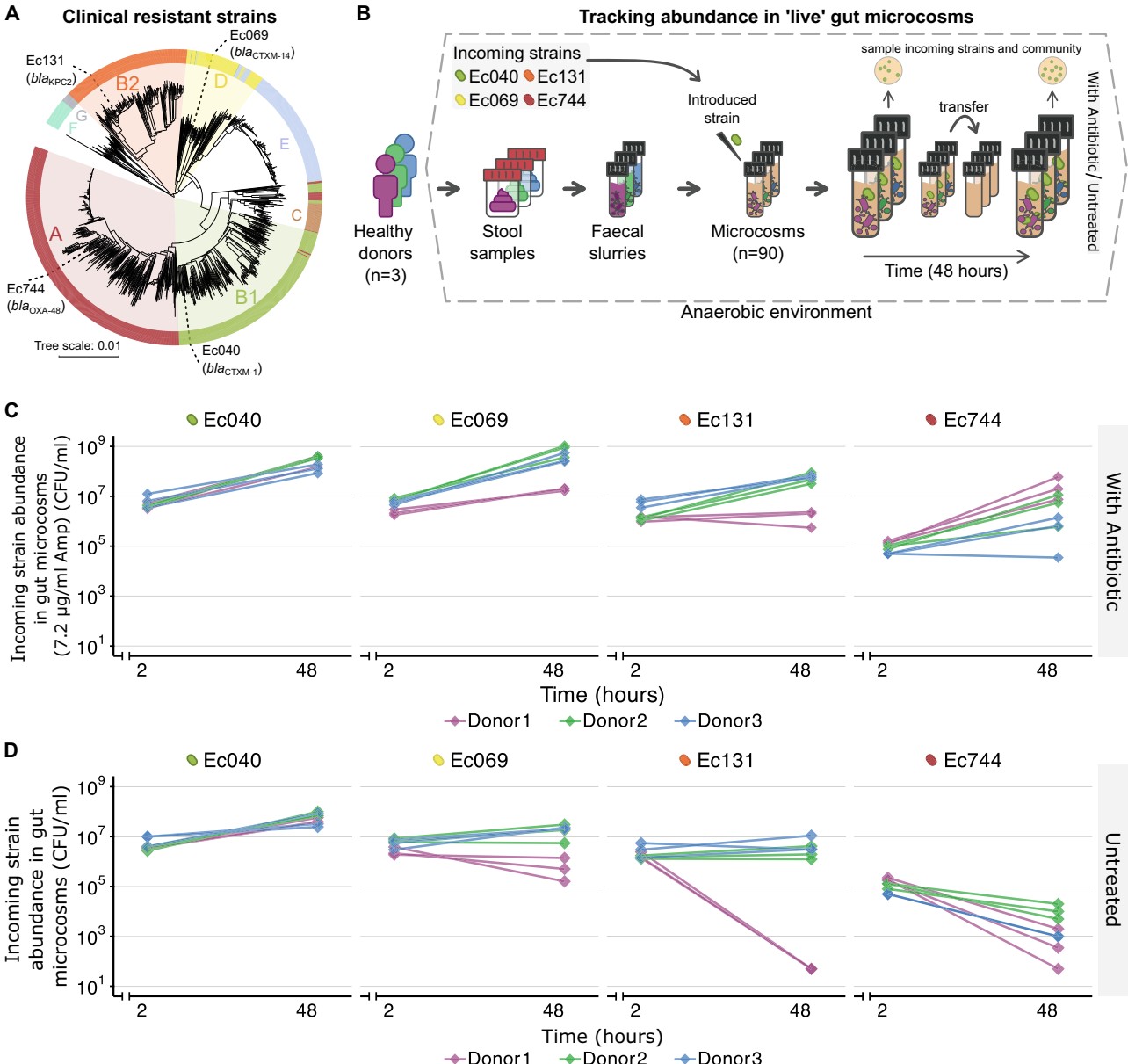

**Fig. 1 | Variable population growth of clinical resistant strains in human 'live' gut microcosms. A** Reference *E. coli* tree (including clinical and non-clinical strains) showing the phylogenetic placement of the four clinical antibiotic-resistant strains used here; strain names used hereafter are given by the outermost labels (see also Supplementary Table 1 & 2). Colours and coloured letters denote phylogroups. Other branches depict 1334 representative reference *E. coli* genomes from the RefSeq database (https://www.ncbi.nlm.nih.gov/assembly), with distances established using Mash v2.0 (see "Methods"). **B** Schematic of the live microcosm assay, where each of the four clinical strains was introduced in microcosms prepared with each of the three stool samples from healthy human donors (see "Methods"). This amounted to 90 microcosms in total (4 focal strains + 1 uninvaded condition × 3 donor samples × with antibiotic/ untreated condition × 3 replicates).

**C, D** Abundance of each incoming antibiotic-resistant *E. coli* strain (labelled at the top of each of the four sub-panels in each plot; further strain information in Supplementary Table 1) over time in anaerobic gut microcosms containing microbiome samples from three human donors (colours; see legend at bottom). The break in the *x*-axis represents the time interval between inoculation (0 hours) and the 2 h sampling point. Three replicate microcosms (lines) are shown for each resistant-strain × human-donor combination, in (**C**) the presence of 7.2 µg/ml ampicillin (Amp) (see "Methods") and (**D**) without antibiotics (untreated). Paired *t* tests showed positive growth over 48 h for all strains with ampicillin ($p < 0.01$). Two-way ANOVAs tested strain, donor, and strain × donor effects on final abundance (see Results section). No pairwise tests are indicated in the figure.

after 48 h > $10^8$ CFU/ml for all four strains; Fig. 2A). This contrasts with the much lower resistant-strain abundances in some combinations in the live microcosm assay (Fig. 1), consistent with resident microbiota inhibiting growth of a laboratory strain observed previously[22]. Nevertheless, as in the live microcosm assay (Fig. 1), final abundances in local abiotic conditions varied among resistant strains (two-way ANOVA, strain effect: $F_{(3,24)} = 16.566$, $p < 0.001$; Fig. 2). This resulted in a strong overall association (across strain-donor combinations) between average incoming-strain abundance in sterilised microcosms and in

the live microcosm assay ($r = 0.84$, $p = 0.001$; Fig. 2B), which was mostly driven by the relatively poor performance of Ec744 compared to other strains in both experiments. A further experiment using sterilised microcosms prepared by a different method (filtration rather than autoclaving) revealed a similar pattern, with strain Ec744 growing relatively poorly (strain effect in two-way ANOVA: $F_{(3,36)} = 56.62$, $p < 0.001$; Fig. 2C and Supplementary Fig. 3A). Strain Ec744 also grew relatively poorly in other relevant abiotic conditions, in the basal medium used to suspend faecal slurries for the microcosm preparation

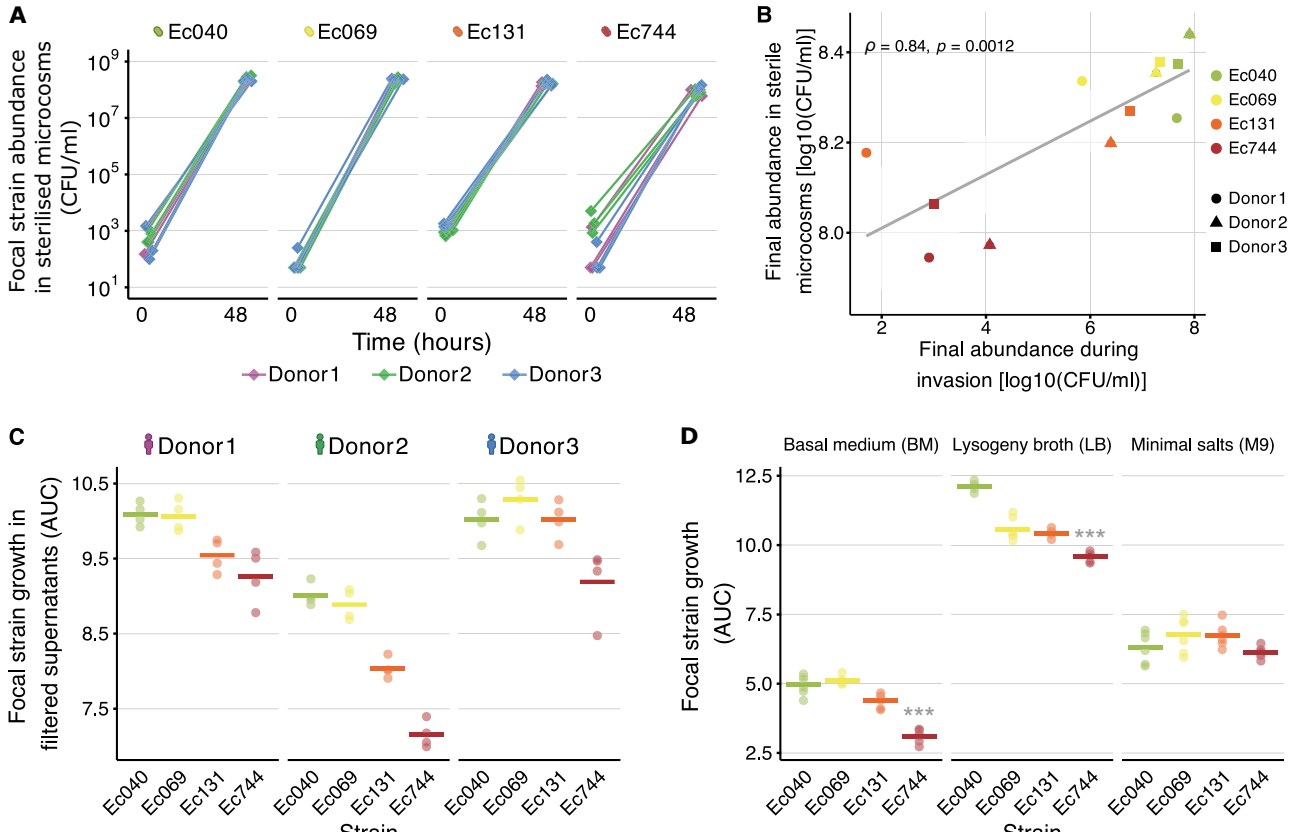

**Fig. 2 | Strain-to-strain variation of population growth in local abiotic conditions. A** Population growth of antibiotic-resistant *E. coli* strains in sterilised microcosms without antibiotics. All strains reached high abundances ( > $10^8$ CFU/ml) after 48 h, but final abundances differed among strains (two-way ANOVA, strain effect: $F(3,24) = 16.57$, $p < 0.001$). Growth of the four resistant *E. coli* strains (labelled at the top of each sub-panel) is shown over time in anaerobic microcosms prepared as in the live microcosm assay, but with sterilised-autoclaved faecal slurry from each of the human donors (colours; see legend at bottom) (See also Supplementary Fig. 2). For each treatment combination, three replicate microcosms are shown. **B** Association between final abundance of incoming strains after incubation in sterilised microcosms (local abiotic conditions) and in 'live' gut microcosms from the 'live' microcosm assay (Fig. 1) (Spearman's $\rho = 0.84$, $p = 0.0012$). Colours/shapes indicate strains and samples (see legend). Each point gives the mean of three

replicates on both axes. **C** Growth performance of each resistant strain (*x*-axis) in microcosms prepared with filtered versions of the faecal slurries from each human donor (labelled at top). Growth performance is estimated as the area under the curve (AUC) from optical density (OD600) data measured over 24 h for each *E. coli* focal strain. Ec744 consistently grew poorly compared to other strains (two-way ANOVA, strain effect: $F(3,36) = 56.62$, $p < 0.001$; see Supplementary Fig. 3A) ($n = 4$ replicates). **D** Growth performance of each resistant strain (*x*-axis) in three types of growth medium (labelled at top), measured as in (**C**) (Supplementary Fig. 3B) ($n = 6$ replicates). Strain effects were significant in BM and LB (one-way ANOVAs, both $p < 0.001$), driven by consistently lower growth of Ec744 (***$p < 0.001$). No significant differences were detected in M9. For **C**, **D** experiments were done in microplates and incubated aerobically. Horizontal lines show medians; points show replicates.

(one-way ANOVA: $F(3,20) = 70.44$, $p < 0.001$; Fig. 2D) and in lysogeny broth (LB; one-way ANOVA: $F(3,20) = 106.8$, $p < 0.001$; Fig. 2D). Across these different conditions, the strain ranking was similar to in our live microcosm assay (Ec040 best on average, Ec744 worst, Ec069 and Ec131 intermediate). By contrast, we detected weaker strain differences in M9 minimal medium plus glucose (one-way ANOVA: $F(3,20) = 2.427$, $p > 0.05$; Fig. 2D). In summary, clinically resistant strains consistently reached high abundances (mean abundance after 48 h > $10^8$ CFU/ml) in microcosms prepared with sterilised microbiome samples compared to those from live samples. Nevertheless, strain performance in relevant abiotic conditions (sterilised microbiome samples) was positively associated with ecological success when the gut microbiome community was present (in the live microcosm assay) (Fig. 2B).

**Strain-specific effects of incoming *E. coli* on resident microbiota**
We next tested for evidence that strain-specific growth performance among resistant *E. coli* strains, both in the presence and absence of antibiotics, was linked to variable interactions with resident microbiota (between-species interactions)[38]. Here, between-species interactions refer to the extent to which introducing a focal strain alters the final abundance of other species in the community, relative to the uninvaded

controls. If such effects vary among introduced strains, we would expect the resulting microbial community composition (estimated by 16S rRNA gene sequencing; Supplementary Figs. 4–7) to vary among gut microcosms inoculated with different resistant strains. In the absence of antibiotics, we detected evidence of this (Fig. 3), with significant differences in community composition depending on the identity of the incoming resistant strain in microcosms with the microbiome sample from Donor3 (PERMANOVA, strain effect for Donor3: $F(3,8) = 1.476$, $p = 0.0357$; Fig. 3A), but not with samples from the other donors (Donor1: $F(3,8) = 0.986$, $p = 0.5145$; Donor2: $F(3,8) = 0.745$, $p = 0.75$; Fig. 3A). Note we excluded the *Escherichia* genus here, so the abundance of the resistant strain itself does not contribute to any observed alteration in community composition. We also found that uninoculated microcosms (controls, not inoculated with any of the four clinically resistant strains, but including resident *E. coli* already present in the microbiome sample) tended to cluster separately from inoculated microcosms (Fig. 3A), showing that introduction of antibiotic-resistant *E. coli* altered the relative abundances of other species.

We identified individual taxa (amplicon sequence variants, ASVs) in the resident microbiota that were either enriched or suppressed in microcosms with an introduced resistant strain compared to control

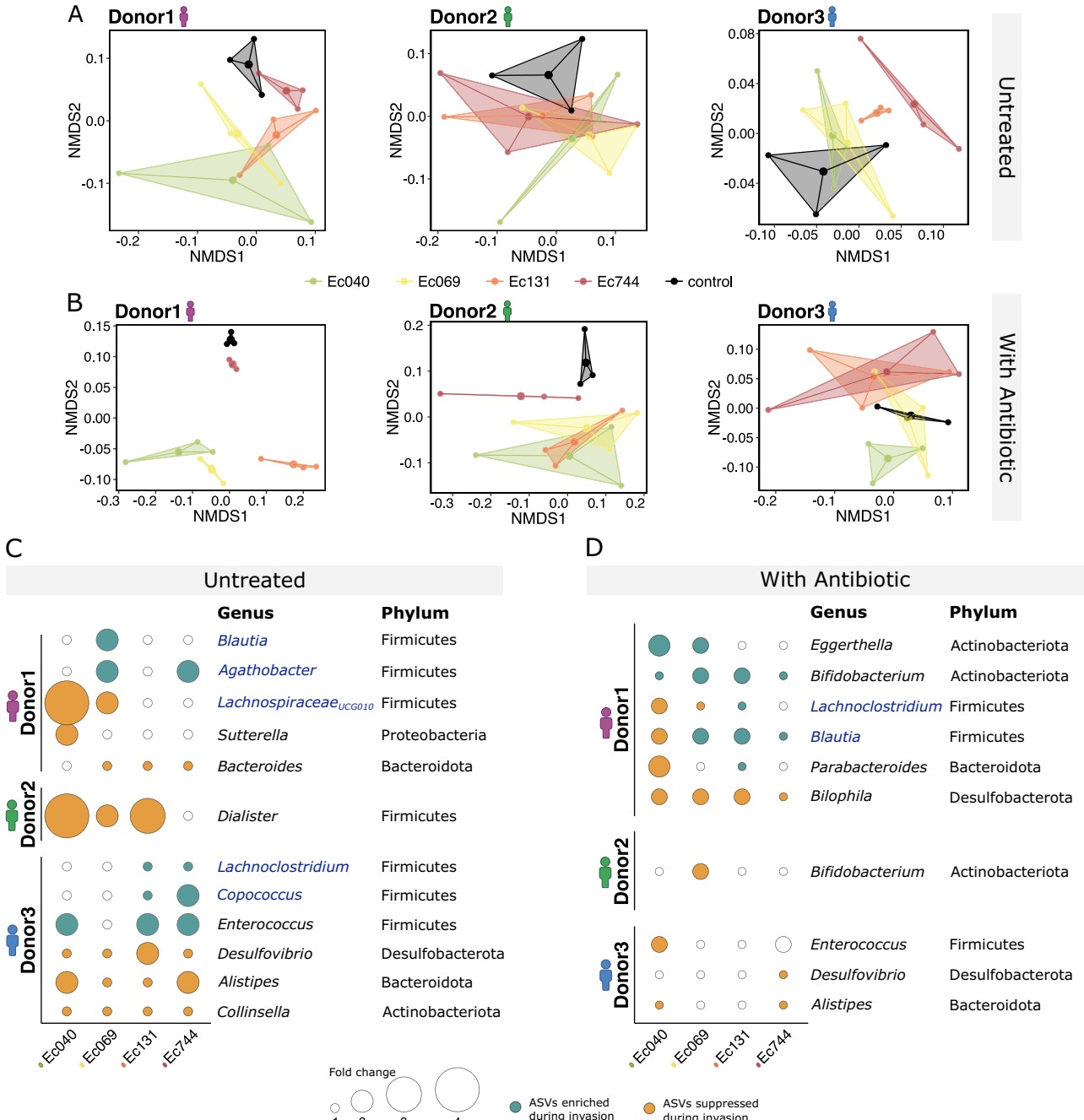

**Fig. 3 | Strain-specific interactions between incoming resistant *E. coli* and resident microbiota in untreated microcosms and treated with antibiotics.** **A**, **B** Non-metric multidimensional scaling (nMDS) of microbial community composition (genus level, based on 16S rRNA gene sequencing) using Bray-Curtis distance matrices, showing all microcosms in groups inoculated with different resistant strains, including an uninoculated control (see legend at bottom of **A**), in the absence of antibiotics for each human donor (panels). PERMANOVA detected a significant strain effect for Donor3 in untreated microcosms (F(3,8) = 1.476, $p = 0.036$), but not for Donor1 or Donor2, and in antibiotic-treated microcosms detected effects of strain (F(3,24) = 2.766, $p < 0.01$), donor (F(2,24) = 107.826, $p < 0.001$), and their interaction (F(6,24) = 2.220, $p < 0.05$). **C**, **D** Linear Discriminant Analysis Effect Size (LEfSe) results showing the amplicon sequence variants (ASVs) enriched or suppressed in inoculated microcosms compared with the uninoculated microcosms (rows), for the set of microcosms with each introduced resistant strain (columns) and from each human donor (groups of rows). Filled circles taxa with an LDA (Linear discriminant analysis) score > 2; circle colour shows the direction of the effect (suppressed/ enriched) and circle size indicates effect size (fold change, see legend). Empty circles (white) show no significant effect. Significance threshold for LEfSe was set at $p < 0.05$. Only taxa showing a significant difference between inoculated and uninoculated microcosms ($p < 0.05$) in at least one strain-donor pair are shown. Genera highlighted in blue belong to the *Lachnospiraceae* family.

uninoculated microcosms (linear discriminant analysis effect size, LEfSe[39]; Fig. 3C and Supplementary Fig. 8B). In the absence of antibiotics, the variability of these effects among incoming resistant strains and donors was largely consistent with the trends detected in the live microcosm assay and consistent in all the replicates. In

microcosms with the microbiome sample from Donor2, we detected relatively few taxa that were impacted by exposure to resistant strains (Fig. 3C and Supplementary Fig. 8B), consistent with the relatively weak impacts of incoming strains in the non-metric multidimensional scaling (nMDS) (Fig. 3A). Similarly, we observed relatively strong

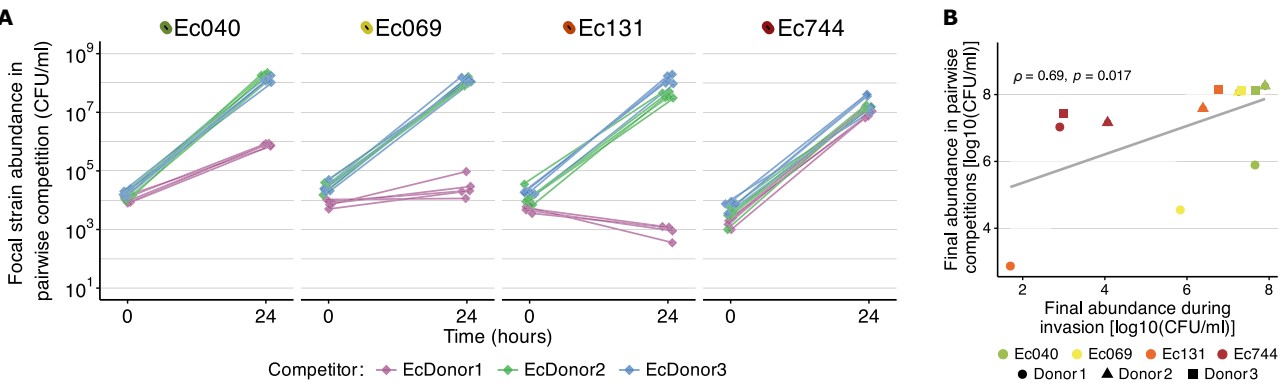

**Fig. 4 | Within-species competition between incoming and resident *E. coli*.**
**A** Population growth of incoming resistant *E. coli* in pairwise competition (co-culture) with resident *E. coli* in sterilised human gut microcosms. Abundance before and after cultivation is shown for each incoming resistant strain (panels), cultured together with each of the three resident *E. coli* strains (colours; see legend), isolated from three donors' microbiome samples. Cultures were incubated in sterilised gut microcosms produced with the corresponding donor stool sample in each combination. Two-way ANOVA on final abundances detected significant effects of incoming strain (F(3,48) = 41.52, $p < 0.001$), donor (F(2,48) = 122.56, $p < 0.001$), and their interaction (F(6,48) = 22.17, $p < 0.001$). **B** Overall association between final abundance of incoming strains in pairwise competitions with resident *E. coli* and during the 'live' microcosm assay (across different strain-donor combinations, see legend at bottom; corresponding to Figs. 4A and 1D, respectively) (Spearman's $\rho = 0.69$, $p = 0.017$). Information about performance relative to the resident *E. coli* in each combination is given in Supplementary Fig. 9.

suppression of other taxa by strain Ec040 (Fig. 3B and Supplementary Fig. 8B), the strongest invader in terms of population growth (Fig. 1), and relatively weak suppression of other taxa by strain Ec744 (Fig. 3C, D), the weakest invader in terms of population growth (Fig. 1). For strain Ec131, LEfSe analysis indicated relatively weak suppression of other taxa in Donor1 microcosms (Fig. 3C, D), consistent with the net negative population growth of this strain in these microcosms (Fig. 1). Across different treatments, in the absence of antibiotics, 58% (7 out of 12) of the taxa where we detected significant impacts of exposure to resistant strains belong to the phylum Firmicutes, with *Lachnospiraceae* being the most altered family (Fig. 3C).

In microcosms treated with antibiotics, we also observed strain-specific impacts on the community depending on the incoming resistant-strain identity (Fig. 3B, D). Final community composition varied depending on the identity of the invading strain (strain effect in PERMANOVA, excluding uninvaded microcosms: F(3,24) = 2.766, $p < 0.01$), and the sample donor (donor effect: F(2,24) = 107.826, $p < 0.001$). Differences among strains depended on the donor (strain × donor interaction: F(6,24) = 2.220, $p < 0.05$), with the greatest differences observed for Donor1 (strain effect tested within each donor: $p < 0.001$ for Donor1, $p > 0.05$ for Donor2 and Donor3). This indicates that even in the presence of antibiotics, the differences among incoming strains drive different changes in specific resident taxa (Fig. 3). We note that while this approach captures interactions that lead to shifts in taxon abundance, it may not capture other forms of interaction, such as neutral or positive associations, that do not result in measurable changes in community composition.

In summary, for the microbiome samples from some of the human donors in our study, introduction of antibiotic-resistant *E. coli* altered community composition in ways that varied among the incoming resistant strains. In the absence of antibiotics, this variability was consistent with differences among resistant strains in terms of intrinsic growth in local abiotic conditions and ecological success in live microbiome samples. For example, strain Ec744 grew relatively poorly in sterilised microcosms (Fig. 2), had negative net population growth in the main experiment (Fig. 1), and had weak effects on other species (Fig. 3).

### Within-species competition between incoming and resident *E. coli*

A further potential contributor to strain-specific ecological success in the absence of antibiotics (Fig. 1) is the variable interaction between incoming and resident *E. coli* strains. The first indication that this may play a role here comes from the relative abundance of *E. coli*, estimated by 16S rRNA gene sequencing at the end of the main experiment (Supplementary Fig. 8). In the absence of intraspecies interaction, population growth of incoming resistant strains and resident *E. coli* would not affect each other[38]. This would lead to a higher absolute abundance of *E. coli* in microcosms inoculated with incoming resistant strains compared to uninoculated microcosms and, all else being equal, a higher relative abundance of *E. coli* compared to other taxa. In some cases, we observed such an increase in relative *E. coli* abundance, but it was inconsistent (Supplementary Fig. 8). In some treatment groups, there was no significant increase in relative *E. coli* abundance (Supplementary Fig. 8B, D), despite incoming strains reaching high abundance (Supplementary Fig. 8A, C) and stable total bacterial abundance (including other taxa, measured by flow cytometry; Supplementary Fig. 8A, C). This suggests that, in at least some microcosms, the introduced antibiotic-resistant *E. coli* suppressed the final density of other, resident *E. coli* strains.

We next tested for variable competitive abilities among the incoming resistant strains in pairwise assays with the most abundant resident *E. coli* strains from each donor sample (EcDonor1, EcDonor2 and EcDonor3, see Supplementary Table 1 and "Methods"). These assays were conducted under the same conditions as the live microcosm assay but in sterilised microcosms, without other resident microbiota (Fig. 4 and Supplementary Fig. 9A). Here, we take competitive ability as the final abundance reached by each incoming strain when co-cultured with a resident *E. coli* strain in sterilised microcosms prepared from each donor sample (Fig. 4; see also Supplementary Fig. 9 for data on performance relative to the resident strain in each combination). This revealed variable competitive abilities depending on both the incoming resistant strain and the donor sample (note the resident *E. coli* strain is different for each donor sample; two-way ANOVA with focal strain final abundance as the response variable, strain effect: F(3,48) = 41.52, $p < 0.001$; donor effect: F(2,48) = 122.56, $p < 0.001$; strain × donor interaction: F(6,48) = 22.17, $p < 0.001$). Across different strain-donor combinations, the final abundance of incoming resistant strains here was positively correlated with their final abundance in the live microcosm assay (Fig. 4B) and with their growth performance in sterilised microcosms (Supplementary Fig. 9C). As in both live and abiotic conditions, this reflected relatively poor growth performance of strain Ec744. Furthermore, cultivation with the resident strain EcDonor1 (in the sterilised Donor 1 sample; Fig. 4)

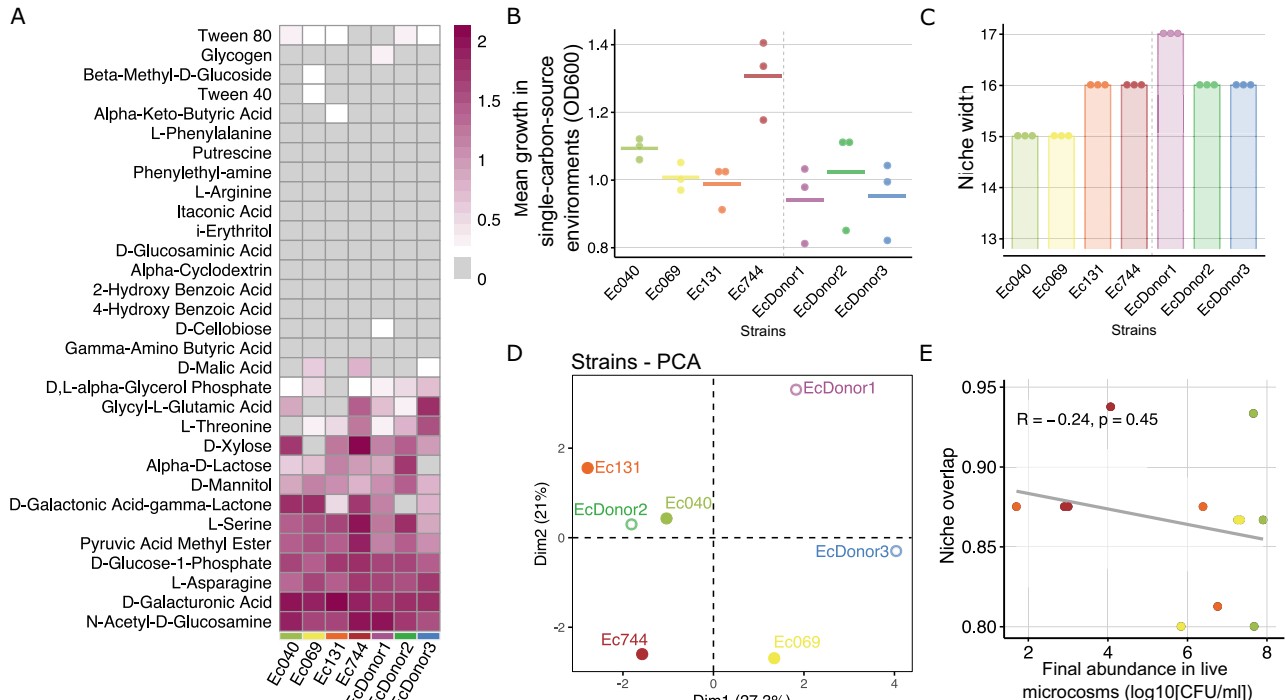

**Fig. 5 | Differences in incoming strains' metabolic profiles were not associated with population growth in live gut microcosms. A** Growth of each incoming resistant strain (Ec040, Ec069, Ec131, Ec744) and three resident *E. coli* strains (EcDonor1, EcDonor2, EcDonor3) across single-carbon-source environments in Biolog EcoPlates. Colour indicates corrected optical density values for each strain (columns) in each single carbon source environment (rows); see scale bar at right. **B** Mean growth of each strain across only those compounds where they showed positive growth (median OD > 0.1 after subtracting the blank); each point shows the mean for one biological replicate (one strain in one EcoPlate, derived from independent overnight cultures); bars show medians. One-way ANOVA detected significant variation among incoming resistant strains (F(3,8) = 12.43, *p* = 0.0022). **C** Niche width for each strain, taken as the number of carbon sources where they

showed positive growth; points show biological replicates (derived from independent overnight cultures). Variation among strains was significant (Kruskal-Wallis test: H(3) = 11, *p* = 0.012). **D** Principal component analysis (PCA) of the *E. coli* strains, determined using growth scores in each single-carbon source environment. Percentages in parentheses next to PC1 and PC2 give the percentage variance explained by each component. **E** Niche overlap between incoming and resident *E. coli* in each strain-microbiome combination, taken as the proportion of carbon sources the incoming strain could use which the resident strain could also use, shown relative to growth performance in live human gut microcosms (mean in each combination as in Fig. 1D). No significant correlations were detected (Pearson's correlation, *p* > 0.05).

supported relatively little growth of some strains, including Ec069 and Ec131, compared to other samples. This is consistent with the relatively low growth observed in 'live' Donor1 gut microcosms (Fig. 1). In contrast, although Ec040 was the strongest invader in terms of population growth in the live microcosm assay (Fig. 1D), it was outcompeted by the resident *E. coli* strain from Donor1 (Fig. 4). This partial overall correspondence suggests that, in addition to within-species interactions, some of the variation in performance in live microcosms may reflect other factors, such as between-species interactions (Fig. 3C, D). In summary, we found evidence that intraspecies competition occurs during invasion by antibiotic-resistant *E. coli*, and this was associated with both their growth performance in local abiotic conditions and their final abundance in microcosms containing resident microbiota. We also detected instances of direct inhibition between strains on agar, but these patterns were not linked to success in the microcosm assays (Supplementary Fig. 10).

## Metabolic profiles of incoming strains do not predict growth success

One possible driver of the observed variation among incoming resistant strains, in terms of intrinsic growth capacity and interactions with other strains and species, is variation of their metabolic phenotype, in terms of their abilities to consume different nutrients[24]. Strains with broader metabolic capabilities, or with unique nutrient-use profiles, might experience reduced competition and therefore grow more successfully across microbiomes[24]. We investigated this by quantifying

the growth profile of each resistant strain across various single-carbon-source environments in Biolog EcoPlates (Fig. 5A and Supplementary Fig. 11). This revealed that the incoming resistant strains varied in their average final population density across different single-carbon-source environments (one-way ANOVA: F(3,8) = 12.43, *p* = 0.0022; Fig. 5B), how many carbon sources they could use (Kruskal-Wallis test: H(3) = 11, *p* = 0.012; Fig. 5C), and which carbon sources they could use (PCA; Fig. 5D). However, we found no association between these properties and population growth success in the main experiment. For example, Ec744 was the worst-performing resistant strain in live microcosms (Fig. 1D), but was not deficient in either which carbon sources it could use or average growth (Fig. 5B, C). This is also consistent with our finding above that strain variation in another single-carbon-source environment, M9 + glucose (Fig. 2D), did not match that in the live microcosm assay.

Competition between incoming and resident bacteria may be more closely predicted by which carbon sources they share[24] (niche overlap) than by how many they can each use. We tested this possibility by estimating niche overlap as the proportion of the incoming strains' usable carbon sources that could also be used by the most abundant resident *E. coli* strains from each microbiome sample (EcDonor1, EcDonor2 and EcDonor3). This revealed no association between niche overlap and either growth performance of incoming resistant strains in pairwise competition with the same resident strains (Pearson's correlation on means from each strain-microbiome combination: *p* > 0.05; Supplementary Fig. 12) or population growth in the

live microcosm assay ($p > 0.05$; Fig. 5E). In summary, growth phenotypes across single-carbon-source environments revealed differences among clinical antibiotic-resistant strains. Unlike growth measurements in sterilised microcosms (Fig. 2), these differences were not associated with strain-specific growth success in human microbiome samples (Fig. 1) showing that among these strains niche overlap is not explaining their growth variability in live microcosms.

## Resident strains can acquire incoming resistance plasmids

The spread of antibiotic resistance in the gut microbiota can result not only from the ecological success of incoming resistant strains but also from the horizontal transfer of resistance plasmids to resident bacteria. To investigate this, we examined whether plasmids could transfer from incoming strains to resident members of the community[10,15]. The four antibiotic-resistance plasmids carried by our invading strains are conjugative[40,41] (Supplementary Fig. 13 and Supplementary Table 2). However, in the live microcosm assay, we detected no transconjugants in the untreated or treated with ampicillin conditions. From 1440 colonies with relevant resistance phenotypes (10 colonies from each of the 144 microcosms with incoming resistant strains; picked after selective plating on chromID ESBL agar for samples with Ec040 and Ec069 and CARBA SMART agar for samples with Ec131 and Ec744; bioMérieux, Switzerland) tested by PCR for both the bacterial host and plasmid identity, all colonies tested positive, indicating they were the plasmid-carrying incoming strain rather than transconjugants. Nevertheless, we do not rule out transconjugants arising that did not form colonies at a detectable frequency during plating from our main experiment. We therefore adapted our experimental design to increase our ability to detect low-frequency transconjugants by using a counter-selectable donor strain, *E. coli* β3914, a diaminopimelic acid (DAP) auxotrophic laboratory mutant of *E. coli* K-12. We carried out a conjugation experiment in anaerobic microcosms, prepared with the microbiome sample from Donor1, with the DAP *E. coli* β3914 as the donor strain with each plasmid, and screening for transconjugants arising from resident strains in the sample by selective plating. Microcosms were supplemented for 24 h with DAP (to allow growth of the donor strain), and then after passage into a fresh microcosm for another 24 h with ampicillin and without DAP (to select for any transconjugants). We identified transconjugants for two of the four plasmids (pESBL15 from Ec040 and pESBL25 from Ec069; Supplementary Fig. 14A). Both strain identity and plasmid carriage were confirmed by PCR. However, the frequency of transconjugants relative to donors was very low in both cases (Supplementary Fig. 14A). Thus, some of the plasmids in our experiment were transferable to resident *E. coli* strains in live human gut microcosms, but the resulting transconjugants did not reach high frequencies even after supplementation with ampicillin.

## Discussion

We found that, upon introduction to healthy-human gut microcosms, the success of clinical antimicrobial-resistant strains depended on multiple layers of ecological interactions, both with and without antibiotics. Antibiotic treatment altered community composition and, in some cases, facilitated the growth of incoming resistant strains, consistent with the potential for competitive release dynamics[42]. Some resistant strains successfully invaded gut microbiome samples even in the absence of antibiotics, while others did not. This strain-specific variability was linked to both intraspecies interactions with resident *E. coli* strains, which in turn were correlated with variable intrinsic growth capacities in relevant abiotic conditions, and to interspecies interactions with other species in the resident microbiota. Together, these results provide new insights into the drivers of invasion success for clinically important resistant strains, including those carrying ESBLs and carbapenemases, and help explain how resistant *E. coli* can displace resident strains even in the absence of antibiotics[16,17].

Our finding that multiple types of ecological interactions are involved in establishment of resistant strains matters, because it advances our knowledge of what needs to be measured in order to predict and manage the spread of resistance. Two examples illustrate this point. First, the observed role of intraspecies interactions suggests comparing resident commensal strains with potentially-colonising resistant strains of the same species can provide valuable insights. This could be done via strain-level phenotyping, such as pure-culture growth experiments in relevant conditions. In our experiments, such data were positively associated with overall performance of incoming resistant strains in live gut microcosms (Figs. 2 and 4). Second, our finding that the incoming resistant strains interact differently with other species in the resident microbiota (Fig. 3), with these effects varying among samples from different individuals, suggests variation of microbiota composition among healthy individuals influences establishment of resistant strains. Additionally, these strain-specific interactions help explain the mechanisms that determine colonisation success. Eventually, this could potentially be accounted for in the management of resistance via metagenomic sequencing of samples from individual patients[34]. A correlation between altered taxonomic composition and susceptibility to colonisation is already established in the context of dysbiosis[43]. Our results suggest a valuable area for future research would be to clarify the connection between taxonomic composition and invasion by specific pathogenic strains in healthy microbiomes. Our data already enhances our understanding in this area. For example, our experimental approach of including microcosms with and without inoculated resistant strains allowed us to demonstrate directly that incoming resistant strains impact the abundance of individual taxa such as the ones belonging to the *Lachnospiraceae* family, considered a protective component of the community[44]. Together, these insights suggest a variety of ecological interactions, beyond single-strain analyses, are relevant for fully understanding and combating antibiotic resistance.

Our findings also shed light on the relevance of different experimental conditions for explaining the variable success of resistant strains invading human microbiomes. We found good agreement between intrinsic growth capacities measured in sterilised microcosms produced with human stool samples and ecological success in live gut microcosms. This is encouraging, suggesting in vitro measurements in pure cultures can predict ecological success of individual strains in complex communities. However, this was not true of multivariate metabolic phenotypes measured across various single-carbon-source environments in Biolog EcoPlates. These and similar types of plates have been used previously to phenotype resistant strains[24,45,46]. These data have relatively high resolution and revealed differences among strains' metabolic fingerprints, but these were not correlated with ecological success in live microcosms. Although the Biolog EcoPlates do not replicate the nutrient conditions of our microcosms, they provide a standardised view of metabolic variation that may be informative in contexts where resource-based competition plays a strong role, or where limiting nutrients can be more precisely identified. Furthermore, we did not see evidence of phages in experiments with filtered supernatant that could explain the differences in population growth in the microcosms, suggesting other factors, such as resource competition or bacterial interactions, are more likely driving the variability in strain performance. Together, these findings suggest that conclusions from experiments aiming to explain or predict the spread of resistant strains, such as assessing the cost of resistance through growth assays relative to susceptible genotypes[47], are most informative when carried out with growth media that come as close as possible to the natural environment of interest. Thus, our results support the employment of urine and faecal extracts in growth assays[47,48]. In addition, the general agreement we saw between sterilised microcosms and results from relevant complex media (LB and

basal medium) indicates appropriate synthetic alternatives, such as artificial urine medium[49], can also be informative.

Although we detected only limited plasmid transfer in our main experiment, our finding that two of the four plasmids could transfer under these conditions highlights that resistance genes may still spread even when the host strain introducing them into a given community fails to grow. Such "source–sink" dynamics, where transient donors fuel gene transfer into better-adapted recipients, described in environmental microbiomes[50], may contribute to resistance maintenance in the gut. Nevertheless, the low frequency of transconjugants we observed shows the importance of strain-plasmid compatibility[15,51] and net population growth for the overall spread of such plasmids.

Our aim was to investigate ecological interactions and their variability among antimicrobial-resistant strains introduced in microbial communities sampled from humans. A key strength of our approach is the use of clinical *E. coli* strains isolated from hospital patients. These strains carry diverse accessory genomes rarely present in lab strains, capturing more realistic interactions with the human gut microbiota. It is important to clarify the scope and limitations. First, the duration of our experiments was focused on the early invasion stages. If the experiments had been longer, evolutionary changes might have played an important role, such as compensatory mutations in resistant strains[52,53]. This would not invalidate our main findings. But the relative importance of different components, such as intraspecific vs interspecific competition, could potentially change in the long term as a result of adaptation to individual microbiomes. Second, the number of strains and samples could be increased. For example, if anaerobic cultivation and accompanying readouts in these conditions could be made much more high-throughput, a greater number of samples and accompanying host metadata could potentially enable associations between specific host factors and susceptibility to colonisation by resistant strains to be drawn. Nevertheless, the scale of our experiments here was sufficient to demonstrate both patterns that were repeatable across different strain-microbiome combinations and key differences attributable to each factor, such as the poor performance of strain Ec744 or the growth success of Ec040 through specific interactions with the resident community. While we worked with a single low antibiotic concentration, higher concentrations would likely trigger stronger disruption of resident communities and alter the balance of microbial interactions. Finally, the physiological mechanisms of interaction between resident and focal strains remain unclear. Despite this, our findings on resource use, pairwise competition in complex media, and direct inhibition in pairwise assays provide some direction for future work in this area, such as further investigating the putative interactions between *E. coli* and *Lachnospiraceae*.

In conclusion, our results provide new perspectives for studying, screening, and designing strategies to combat antimicrobial resistance. Our findings demonstrate the critical role of intraspecies competition within *E. coli* populations, revealing that this dynamic significantly influences the persistence and spread of resistant strains. Beyond resistance, this suggests that intraspecies competition could play a crucial role in the clonal turnover observed in longitudinally sampled individuals[54]. Our other findings indicate there is an additional layer, in that different incoming resistant strains also interact with other species, altering the taxonomic composition of the resident community, potentially driving the outcome of their growth success. This effect varied among samples from different healthy individuals, revealing an important implication of inter-microbiome variability, which has been widely documented in recent decades[55]. Finally, by showing that the growth of individual strains in live microcosms was linked to their performance in relevant abiotic conditions and pairwise cultures with competing resident strains, our results can inform experimental methodology aiming to identify high-risk strains or strain-microbiome combinations.

## Methods

### Bacteria and growth conditions

We used four clinical *E. coli* strains as focal strains (Supplementary Table 1). These were obtained from hospitalised patients in two different studies at the University Hospital Basel, Switzerland[56,57]. The healthy donors were sampled in Zürich, approximately 90 km from Basel, meaning the resistant strains used here are epidemiologically relevant to the region and could plausibly be encountered by the studied donor populations. The four strains belong to four different phylogroups and sequence types (STs), and carry different conjugative antibiotic-resistance plasmids (Supplementary Table 1 and 2). Ec040 and Ec069 carry pESBL15 (IncI, 88.9 kb) and pESBL25 (IncFIA, IncFIB, 131 kb), respectively, encoding extended-spectrum beta-lactamases (ESBL) of the CTX-M type. Ec131 and Ec744 carry pKPC and pOXA-48, respectively, with resistance genes encoding carbapenemases, a subgroup of beta-lactamases. All four strains are resistant to ampicillin (MIC > 512 mg/L). We cultivated bacteria in lysogeny broth (LB) or in M9 minimal medium supplemented with 0.8% glucose and 1 mM $MgSO_4$. To test intraspecies competition within the resident community, we used representative resident *E. coli* strains isolated from the three microbiome samples from our main experiments (EcDonor1, EcDonor2, EcDonor3; see Supplementary Table 1). These resident strains were isolated as described previously[58] by plating faecal slurry on chromatic agar (Chromatic MH, Liofilchem, Roseto degli Abruzzi, Italy), picking 10 colonies from each donor slurry and testing them by Repetitive extragenic palindromic (REP) PCR[59]. All the tested colonies from each donor sample belonged to the same strain and were different among the donors (Supplementary Table 1). This project and the one described in Boumasmoud et al.[58] were developed in parallel in the same laboratory using the same sampling collection, but addressed distinct research questions. Specifically, the present study focuses on clinical antibiotic-resistant *E. coli* strains isolated from patients, and their invasion dynamics and impact on human gut microbiota, topics not explored in the other work.

### Microbiome samples from healthy humans and microcosm preparation

Stool samples were collected from healthy subjects at the Department of Environmental Systems Science, ETH Zürich, Switzerland, on 17 March 2022 and 24 May 2022 (approved by the Ethics Commission of ETH Zürich, number EK-2020-N-150). Written informed consent was obtained from all participants before sample collection. Inclusion criteria were: over 18 years old, healthy, not obese, not recovering from surgery, not taken antibiotics in the past six months, and not tested positive for SARS-CoV-2 in the past two months. Each sample was collected in a 500 ml plastic specimen container (Sigma-Aldrich) and kept anaerobic using one AnaeroGen anaerobic sachet (Thermo Scientific, Basel, Switzerland). The three samples used for the experiment were randomly selected from a larger number of donated samples. These stool samples are a subset of the whole collection, where samples from Donor1, Donor2, and Donor3 have the alternative IDs M2, M3, and M4 in Boumasmoud et al. (2024). We collected samples before the live microcosm assay and kept them for maximum 1 h before processing to start the experiment. From this step, all experiments using faecal samples were done in anaerobic conditions in a vinyl anaerobic chamber (Coy, USA). To prepare faecal slurry from each sample, we suspended 20 g of sample in 200 ml anaerobic peptone wash (1 g/l peptone, 0.5 g/l L-Cysteine, 0.5 g/vwl bile salts, and 0.001 g/l Resazurin; Sigma-Aldrich) to prepare a 10% (w/v) faecal slurry. We stirred each slurry for 15 min on a magnetic stirrer to homogenise, followed by 10 min of resting to sediment. At this point, we removed 100 ml of each faecal slurry ('fresh slurry'), which we used later to reintroduce the resident microbial community to sterilised slurry (for the live microcosm treatments). To sterilise faecal slurries, we transferred 100 ml to a 250 ml Schott bottle and autoclaved for 20 min at

121 °C. We then prepared each microcosm using faecal slurry from one of the three donor samples (350 µl fresh slurry plus 500 µl sterilised slurry) plus 7.5 ml basal medium (2 g/l Peptone, 2 g/l Tryp- tone, 2 g/l Yeast extract, 0.1 g/l NaCl, 0.04 g $K_2HPO_4$, 0.04 g/l $KH_2PO_4$, 0.01 g/l $MgSO_4x7H2O$, 0.01 g/l $CaCl_2x6H_2O$, 2 g/l $NaHCO_3$, 2 ml Tween 80, 0.005 g/l Hemin, 0.5 g/l L-Cysteine, 0.5 g/ l Bile salts, 2 g/l Starch, 1.5 g/l Casein, 0.001 g/l Resazurin, pH adjusted to 7.0, addition of 0.001 g/l Menadion after autoclaving; Sigma-Aldrich)[22].

### Tracking abundance in human gut microcosms (live microcosm assay)

To prepare anaerobic cultures of each focal strain prior to inoculation into microcosms, we streaked each focal strain on LB agar (Sigma-Aldrich, Buchs, Switzerland), incubated overnight, then picked 72 randomly-selected colonies (18 per focal strain) into 72 Hungate tubes (VWR, Schlieren, Switzerland) containing anaerobic LB (Sigma-Aldrich), and incubated at 37 °C overnight. We then used these 72 independent cultures to inoculate 8 µl of one focal *E. coli* strain (approximately 1:1000 *v:v* dilution) per microcosm. We included three replicate microcosms in each combination of human Donor (1, 2 or 3), focal strain (present or absent), and antibiotic (treated or untreated). The total number of microcosms was 90 (4 focal strains + 1 uninvaded condition × 3 samples from healthy donors × with antibiotic/ untreated condition × 3 replicates). In the antibiotic treatment, we added ampi- cillin to a final concentration of 7.2 µg/ml, approximating the 90% inhibitory concentration (IC90) for the laboratory reference strain *E. coli* MG1655[22]; this was introduced 2 h after the focal strain, right after sampling microcosms and storing aliquots for this initial timepoint. We incubated microcosms at 37 °C anaerobically without shaking. After 24 h, we transferred 800 µl from each microcosm to a new microcosm (containing basal medium plus 500 µl of sterile slurry from the cor- responding human donor, supplemented with ampicillin in the anti- biotic treatment), before sampling microcosms and storing aliquots (with 50% glycerol at − 80 °C) after 48 h.

To estimate the abundance of the incoming strains at the start and end of the experiment, we used a combination of selective plating and colony PCR. The plasmid carried by each focal strain was used to identify the focal strain over time. For selective plating, we serially diluted samples from each microcosm and plated them on chromo- genic medium chromID agar -ESBL and CARBA SMART- (Biomerieux, Switzerland), which are specific for the resistance phenotypes carried by the plasmids in our incoming resistant strains. In preliminary screening of each donor sample (slurry) using chromogenic medium chromID agar -ESBL and CARBA SMART -, we found no colony-forming units of resident bacteria on these types of plates. As a further con- firmation step, we used colony PCR to confirm strain identity after plating for the incoming strains, testing 10 colonies from each of 144 microcosms (4 focal strains × 3 donors × 3 replicates × 2 antibiotic conditions × 2 timepoints). The reaction mix consisted of 2 × GoTaq green master mix, 2.5 µM of each primer, and nuclease-free water. The thermal cycle programme ran on a labcycler (Sensoquest, Göttingen, Germany) with 6 min 95 °C initial denaturation and 30 cycles of 95 °C for 1 min, 58 °C for 30 s, 72 °C for 35 s, and a final elongation step of 72 °C for 5 min. For gel electrophoresis, we transferred 5 µl of the PCR reaction to a 1.5% agarose gel stained with SYBR Safe (Invitrogen, Thermo F. Scientific) and visualised by UV illumination. We also tested for evidence of plasmid loss during cultivation in our experimental conditions (in sterilised microcosms; see below); we detected very similar numbers of colonies on selective vs non-selective agar after 48 h incubation, indicating plasmids were stable in all combinations (Supplementary Fig. 15).

To estimate total microbial abundance in each microcosm, we used flow cytometry. Briefly, we diluted samples by 1:10,000 with sterile and filtered phosphate-buffered saline (PBS) and stained them with SYBR Green (Invitrogen, Thermo Fisher Scientific). We used a Novocyte 2000R (ACEA Biosciences, San Diego, CA, USA), equipped with a 488 nm laser and the same filter setup as in previous work[22].

### Experiments in bacteria-free microcosms

We measured the growth of the four incoming strains in sterilised microcosms using two different methods. First, we prepared sterile versions as for live microcosms, but with only sterilised (autoclaved) faecal slurry (850 µl). These autoclaved versions were tested in LB agar plates for bacterial growth. Then, they were inoculated with the incoming strains as in the live microcosm assay (seen in Fig. 1B), but with a lower inoculum size, and incubated anaerobically as above. The reduced inoculum allowed for a longer growth phase, increasing the assay's sensitivity to detect growth differences between strains. Second, we prepared filter-sterilised versions of each faecal slurry by transfer- ring 3 ml to a 15 ml tube and centrifuging (10 000 r.p.m., 5 min), before syringe-filtering (0.22 µm pore). These were inoculated with focal strains by 1:1000 dilution in 96-well microplates, then incubated aero- bically for 22 h at 37 °C with shaking (250 r.p.m.) in a plate reader (Tecan NanoQuant Infinite M200 Pro), measuring optical density (OD600) every 15 min. From these data, we extracted the maximum growth rate (µ), maximum optical density (ODmax), and area under the curve (AUC) for each well using growth rates and flux packages in R. Both autoclaved and filtered versions were tested in LB agar plates for bacterial growth. We performed further growth assays, as in filtered slurry, in three types of slurry-free growth medium: the basal medium used to prepare microcosms in the live microcosm assay, LB, and M9 minimal medium supplemented with 0.8% glucose and 1 mM $MgSO_4$.

### Inferring microbial community composition by 16S rRNA gene sequencing

We thawed samples of fresh faecal slurry and samples from each microcosm at the end of the live microcosm assay on ice, then homogenised by vortexing. We then extracted DNA following the protocol of the DNeasy PowerLyzer PowerSoil Kit (Qiagen). DNA yield and quality were checked by Qubit and Nanodrop. We amplified the V3 and V4 region of the 16S rRNA gene with three slightly modified uni- versal primers[60] with an increment of a 1-nt frameshift in each primer to increase MiSeq sequencing output performance between the target region and Illumina adaptor. The target region was amplified using 4 primer pairs in a limited 17-cycle PCR, and performing 4 different PCR reactions per each primer pair combination separately. We then pooled the resulting PCR products from the four PCR reactions in equal volumes, resulting in a total volume of 90 µl per sample. We cleaned up PCR products using AMPure XP Beads, and in a second PCR, attached adaptors with the Illumina barcodes of the Nextera XT index Kit v2. We normalised quantified samples and pooled them in equimolar amounts. Before to load the library, we performed a final cleanup in order to remove additional fragments. We checked the fragment size on the Agilent 2200 TapeStation and quantify the library with the Qubit Fluorometer (Invitrogen) and qPCR using the KAPA library quantification Kit (KAPA Biosystems. Wilmington, MA, USA) on the LightCycler 480 (Roche, Basel, Switzerland). We then finally load the library with 10% PhiX on the Illumina MiSeq platform with the Reagent Kit V3 at the Genetic Diversity Centre, ETH Zürich.

Resulting raw data was demultiplexed and trimmed for Illumina adaptor residuals and locus-specific primers. Analysis was performed with QIIME2 (2021.4)[61]. To obtain amplicon sequence variants, reads were processed with the Divisive Amplicon Denoising Algorithm (DADA2)[62] with a quality score threshold set at 25. The truncation length was set to 200 for forward and reverse reads. Bacterial tax- onomy was assigned using SILVA v138 99% 16S full-length database. The first step filtered out features classified as mitochondria and chloroplasts. Amplicon sequence variants were used to generate the tree for the phylogenetic diversity analysis with FastTree2[63] (Fig. 1A). To calculate alpha-diversity indices (observed features, Shannon and

Faith's phylogenetic diversity), beta-diversity indices (Bray-Curtis distance and weighted/ unweighted UniFrac) and to perform nMDS, we used the q2-diversity pipeline. To test for treatment group effects on community composition, we used permutational multivariate analysis of variance (PERMANOVA). To identify ASVs with differential abundances in the microcosms after being invaded by each of the four focal strains, we utilised the LDA Effect Size (LEfSe) analysis[39]. This method can be used to compare the statistically significant differences between groups (linear discriminant analysis (LDA) > 2).

### Testing for pairwise interactions among incoming and resident *E. coli*

For pairwise competition assays in sterilised microcosms, we inoculated sterilised autoclaved microcosms prepared as above (see Experiments in bacteria-free microcosms) with each of the incoming resistant strains and each of the resident *E. coli* strains (see Bacteria and Growth Conditions above), in pairwise combinations in the absence of ampicillin. For each combination (incoming and resident strain), we made four replicates. Following the same procedure as before (see Experiments in bacteria-free microcosms), we inoculated each pair of strains in a 1:1 *v:v* ratio. After 24 h at 37 °C in anaerobic conditions, we made serial dilutions and plated on selective agar (only resistant strains) and LB agar (both strains). Using abundances of each strain at each time point, we estimated the selection-rate constant in each assay[64]. In these assays, it would be possible for any transconjugants that arise during the assay to falsely inflate our estimate of resistant-strain abundances. This is unlikely to have contributed to our conclusions, because in separate conjugation assays with a similar culturing set-up, transconjugants only appeared in two of four combinations, and only reached very low frequencies (Supplementary Fig. 15A).

As a second test for interactions among *E. coli* strains, we used an agar overlay assay. For each assay, we prepared a soft-agar overlay inoculated with one strain (0.5% Agar in LB broth), before adding a droplet (10 μl) of a second strain, and incubating at 37 °C for 24 h. For each combination of resident and incoming resistant *E. coli*, we tested for inhibition in both directions (with resident *E. coli* in the overlay and incoming resistant *E. coli* in the droplet, and vice versa). We visually inspected each plate after incubation for evidence of inhibition, such as halo effects.

### Measuring growth across single-carbon-source environments

Utilisation of carbon sources by incoming strains and resident strains was assessed using Biolog EcoPlates (Biolog, Hayward, California) following the manufacturer's protocol. Briefly, strains were grown in LB overnight and then diluted in phosphate-buffered saline (PBS) for incubation in the Biolog EcoPlate. The EcoPlate includes three replicated sets of wells, each containing one of 31 organic carbon substrates or water (blank), plus a redox-sensitive tetrazolium dye. We tested each strain in three sets of EcoPlate substrates, distributing each replicated set of assays (32 wells) for each strain randomly across the different Biolog plates (that is, we accounted for plate effects by testing each strain across multiple plates). Each EcoPlate was then incubated at 37 °C for 24 h, before taking OD measurements at 590 nm in a Tecan plate reader. We corrected OD measurements by subtracting the value for the corresponding blank well (control well in each replicate set of wells). We used a cut-off value to assign carbon sources as being used (positive impact on growth) or not by a given strain; we defined the cut-off as median OD600 nm after subtracting the blank > 0.1 (Supplementary Fig. 11). To measure niche overlap between invading and resident strains, we calculated the proportion of carbon sources that the invading strain can use that were also used by each resident strain.

### Testing for evidence of conjugative plasmid transfer

To test for transconjugants during the live microcosm assay, we used PCR screening of 1440 colonies (10 each from 144 invaded microcosms), picked after plating on chromogenic medium (chromID ESBL -for the samples with Ec040 and Ec069- and CARBA SMART -for the samples with Ec131 and Ec744- agar; bioMérieux, Switzerland) to select for the plasmids. All tested colonies were identified as *E. coli* and as the same strain as the incoming resistant strain, indicating they were the plasmid-carrying incoming strain rather than transconjugants.

We also made three further tests for horizontal transferability of the focal plasmids. First, we used a mating assay between the focal strains (native clinical strains) carrying the plasmids (plasmid donors) and a chloramphenicol-resistant *E. coli* K-12 MG1655 (CmR, ΔgalK::cat; recipient for the plasmid). We did this assay on LB agar and in LB broth (Supplementary Fig. 13A). Second, we tested the three resident *E. coli* strains as potential recipients for the plasmid (Supplementary Fig. 13B). Third, we tested for the emergence of transconjugants in conditions closer to those of the live microcosm assay, but with a lower detection limit, by using counter-selectable *E. coli* β3914 transconjugants as plasmid donors, inoculating them into microcosms prepared as in the main experiment, and testing for the emergence of transconjugants from resident *E. coli* in each sample. We used the same protocol as in other experiments to isolate transconjugants (Supplementary Figs. 13 and 14), but during the first 24 hours we added DAP 0.3 mM, to allow the donor strain to grow in the presence of the resident microbial community. After 24 h, we propagated the community in a fresh microcosm without DAP, but with ampicillin at the same concentration as in the antibiotic treatment of the live microcosm assay, to select for any transconjugants arising from resident bacteria. The presence of each plasmid and strain identity were checked by PCR (Supplementary Table 3).

### Statistics & reproducibility

No statistical method was used to predetermine sample size. No data were excluded from the analyses. The experiments were not randomised. The investigators were not blinded to allocation during experiments and outcome assessment. All experiments were conducted with at least three biological replicates, as indicated in the relevant figure legends. Replicates were defined as independent cultures or microcosms derived from distinct donors or strain preparations. The reproducibility of the results was confirmed across independent experiments and donor microbiomes.

### Whole genome sequence analysis

To determine the distribution of the *E. coli* isolates across the phylogeny of the species, we obtained 1334 assemblies of *E. coli* complete genomes from the RefSeq database (https://www.ncbi.nlm.nih.gov/assembly). Distances between genomes were established using Mash v2.0[65] and a phylogeny was constructed with mashtree v0.33[66]. The tree was represented with a midpoint root using the phytools package in R[67] and visualised using the iTOL tool[68]. To determine the phylotype of each genome, we used the ClermonTyping tool[69].

### Ethics statement

This study complies with all relevant ethical regulations. The sampling protocol was approved by the Ethics Commission of ETH Zürich (approval number EK-2020-N-150). Written informed consent was obtained from all human participants prior to sample collection.

### Reporting summary

Further information on research design is available in the Nature Portfolio Reporting Summary linked to this article.

## Data availability

Sequences of the 16S rRNA amplicons from the 90 microbiome samples and the genomes of the three resident *E. coli* strains are uploaded to the NCBI and are accessible through projects PRJNA1179737 and PRJEB80311. All Source Data supporting the results of this study is

provided with this paper and available through Figshare under the following https://doi.org/10.6084/m9.figshare.29669927. Source data are provided in this paper.

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

## Acknowledgements

This work was supported by the Swiss National Science Foundation (SNSF project 310030_192428; A.R.H.).

## Author contributions

R.L.-S. and A.R.H. conceived this study. R.L.-S. performed the experimental work with help from M.B., K.P.-C., and M.R. R.L-S. carried out the analysis of experimental results and performed the computational analyses. A.E. and N.N. contributed materials, strains and data from clinical strains. R.L.-S. and A.R.H. coordinated the study. R.L.-S. and A.R.H. wrote the initial draft of the manuscript, and all the authors contributed to the final version of the manuscript and approved it.

## Competing interests

The authors declare no competing interests.
