## [Transparent Peer Review file · Nature Communications]

Multi-layered ecological interactions determine growth of clinical antibiotic-resistant strains within human microbiomes

Corresponding Author: Dr Ricardo Leon-Sampedro

Version 0:

Reviewer comments:

Reviewer #1

(Remarks to the Author)

This manuscript by Leon-Sampedro et al., addresses the ability of clinically relevant, antibiotic-resistant *E. coli* strains to invade the intestinal microbiota of healthy individuals. The authors report on differences in growth of the *E. coli* strains within microcosms of healthy donors' intestinal microbiota. Further tests reveal associations between this phenotype with monoculture growth and competition with resident *E. coli* of the intestinal microbiota.

What stands out to me is the very elegant experimental set-up and that is close to the clinic. Lots of work has been done and the individual experiments are sound. I do think that the authors are asking the right questions and I strongly believe that meaningful conclusions can be drawn from the work on this set up in the future. At this moment, I am afraid that I had difficulty in connecting some of the dots, as described in more detail below.

Major concerns

1. Conceptual framework. When starting to read this manuscript, I thought that one unique angle of this study would be the focus on antibiotic resistant *E. coli* isolates. Already after the first section of the results, it turns out that the statistically significant effect is observed in the absence of antibiotics and from this point onwards, it feels a bit as the antibiotics do not really matter anymore for the story and very similar observations could have been made by asking how a non-resistant *E. coli* strain, maybe even a lab strain, would colonize these microcosms. To be honest, I think such a simpler set up would already be challenging enough to disentangle this in a meaningful manner.
2. Distinction from other ongoing work. The point above becomes even more severe in light of reference 48. I appreciate that the authors bring up their other work. As I understand, the angle of using antibiotic-resistant *E. coli* is one point that distinguishes this from the other work. If the angle of antibiotics does not really matter much here, maybe the two stories are not that distinct any more or the story here would need to focus more on the antibiotics again to become distinct more clearly.
3. Sample size and generalizable conclusions. I am wondering whether the current version of the manuscript tries to achieve too much with too small of a sample size. It is just really many variables and therefore I think hard to find meaningful associations or generalizable principles (different *E. coli* strains with different plasmids going into different healthy donors). Maybe one way out would be to keep the set up and sample size and focus more on one individual strain or donor or resistance plasmid and go more in depth and potentially try to test causation of one of the observed correlations or go for a more mechanistic understanding.

Minor comments

For multiple figures, please make sure outside ticks are added to all y-axes, it is not always clear what the numbers on the y-axis refer to. Especially in Fig. 2b, c, it is not entirely clear.

It might be worth rethinking which figures should go in the main text and which ones in the supplement, e.g. if the aim from fig. 2 onwards is to find associations/correlations with the phenotypes observed in the first figure, wouldn't it help to include

the data on these associations in the respective subsequent figures (e.g. move Fig. S2 to the main Fig. 2) rather than having it in the supplement?

L. 183, use of the term “interaction”, I am aware that this could become a bit philosophical. Is it truly an “interaction” to see the addition of one strain go along with the composition of the community composition in the slurry?

I. 73, for clarification, it might arise elsewhere, where did the four clinical strains come from? E.g. from patients in the same area than the healthy individuals? Is there a real chance of the healthy individuals in this study acquiring these resistant strains?

Fig. 1

- panel A, the label of the four strains is not entirely clear to me, does the end of the black line indicate where they are in the tree or rather which phylogroup they belong to. I am also not clear about the heading “clinical resistant strains”, are all strain in the tree clinically relevant or only the four in the focus on this study?
- panel B, is it really n=90 microcosms? I do not fully understand the number or the base on which it was calculated.
- What was the rationale for this exact concentration of 72.ug/ml? It might be worth to discuss the implications of the antibiotic on the microbiota per se more in the later sections.
- I. 104 “In the presence of ampicillin, all four introduced resistant strains underwent net positive population growth over 48h (Fig. 1C).” Is this backed up by statistics?

Fig. 2

- Fig. 2b, what is the number of replicates and what does the horizontal line show? The figure legend says median, to me it looks like the mean.
- Fig. 2b/c, what is the number of replicates here? n=4? Would be good to add this to the figure legend.

Fig. 4,

- is there data on the competing strain? Might be good to write out based on which exact data or kind of data the term “competition” is being used.
- what was the rationale of doing 5 replicates here (and 3 replicates for figure 1)? I do think that the analyses of these experiments will be very sensitive to the number of replicates. If 5 replicates would be done in Fig. 1, I could imagine a modified outcome of the statistics done there. In general, I think the more replicates the better.

Fig. 5

- I. 288, what was the authors’ rationale to look into carbon sources? was it based on existing literature or based on existing data on their set up that carbon sources would be limited. Wouldn’t we only expect to see a correlation here if carbon sources would be of limited supply in the experimental set up?

I am making this point here, I think it applies also a bit more generally that the associations that are being tested for do come a bit out of the blue and seem driven a lot by practical reasons (e.g. available biolog system) rather than careful consideration of what competing resources the individual E. coli strains could face in the very specific system that is under study.

- Fig. S13/l. 329 ff I do agree that this is interesting, it seems a bit disconnected though. E.g. the inhibition phenotype was observed on plates and the microcosms experiments were done in liquid. Based on this difference alone, there could be many reasons why there is no association.

- Fig. S14/S15, l. 340 ff I see the point and the work and I think crucial to get to back to the original research question again. In my view, it somehow does not help the story much in its current form and rather seems like an add on, maybe because somebody asked for this, rather than a possibility that was taken into account from the beginning or was in mind from the start. In general, I like the set-up of using a lab strain with the different plasmids because it goes in the direction of disentangling which of the many variables (plasmid, strain, host community) in the original set up do play a role and make a difference. As I wrote at the beginning, I think there is tremendous potential in the experimental set up and as exciting as it might be to address all these questions that are raised directly or more indirectly in this manuscript, it might be easier for this reader if there would be a clear story line that addresses one or fewer questions more consistently.

Reviewer #2

(Remarks to the Author)

In this manuscript, the authors track the ability of four clinically isolated *Escherichia coli* (Ec) harbouring antibiotic resistance plasmids to invade human faecal microbial communities in vitro. The authors hypothesise that different strains will have different abilities to invade resident communities, likely due to differing intrinsic growth abilities, interspecific interactions or niche overlap.

The authors introduce the focal Ec strains into communities with and without positive selection for the focal MDR Ec strains (ampicillin treatment). In the presence of positive selection, all of the focal strains are able to maintain positive growth and invade the community, however in its absence the results are variable, with some strains unable to become established. The leading result is that the intrinsic growth ability of the strains (i.e. their growth rate in the absence of the communities) is highly correlated with their ability to invade the community. In contrast, niche overlap and interference competition with other

resident Ecs are poor predictors of the growth ability of the focal strains in the presence of the communities.

While the presented work does advance our understanding of the criteria that enable resistant pathogenic bacteria to become established within complex multispecies communities, I have some concerns that require addressing in the current manuscript.

Major comments:

1. It is unclear to me why are the inoculum densities so different between the sterile (figure 2a) and live (figure 1c/d) microcosms. Are the focal strains unable to invade if inoculated at low density? Can these experiments be directly compared given the $\sim 10,000\times$ difference in inoculum density ($\sim 10^3$ vs $\sim 10^7$)?
2. Given the very high inoculum density in the live microcosms who is invading who? It is unclear what the initial density of the faecal community is at T0 (final densities are reported in S9). But presumably this very high inoculum density is much higher than any one individual strain/species within the mixed faecal communities.
3. In the absence of antibiotic selection is there space for the focal Ec strains to expand into or are the populations already at carrying capacity at T0? This may explain why there is net positive growth in the addition of antibiotics. Ab treatment is likely killing a substantial part of the resident population allowing the focal Ec to expand into, whereas in the absence net growth is close to zero (or below) for strain/community combinations. This may help to explain why Ec744 is unable to be maintained in the absence of selection, given its relatively low
4. The authors state that interactions can be deduced from differential changes in the relative abundance of members of the resident community. While I agree that this is true for some interactions, others such as commensalism (+/0) and amensalism (-/0) would not result in changes in the relative abundance of community members. Therefore, you cannot rule out interactions playing an important effect simply because the community composition does not change. This may explain why significant changes in community structure were not observed in Donor1 despite it having an inhibitory effect on three out of the four focal species.

Minor comments:

1. Line 28-29: "success was positively associated with... their competitive interactions with resident E. coli." The data does not convincingly show this. In fact, the authors state on line 333, "the patterns of inhibition were not closely associated with growth success in the live microcosm system".
2. Presumably ampicillin resistance is provided by the resistance plasmids that the focal Ec's carry, this is not made completely clear at the start of the results.
3. The phrase "more differently" on line 115 is vague; consider revising.
4. Figure S1 – the use of box plots with three data points in this figure seems redundant as the data points just fall on the whiskers and median line.
5. Figure S2 – I suspect you would see a very similar positive correlation between the final abundances of the sterile cultures and the antibiotic-treated cultures because the carrying capacity of Ec744 strain is lower than the other strains.
6. Line 90-92: I appreciate that the authors use clinically isolated strains but they justify their use by stating, "the use of clinical strains and plasmids accounts for possible genomic factors that may influence colonisation success, such as pathogenicity islands, prophages or other plasmids, which are often not represented in domesticated or model strains." But how does the use of four strains account for this? The role of genomic differences was not examined, and there likely isn't enough data to conclude how such specific genomic differences between the strains alter the outcomes presented here.

Version 1:

Reviewer comments:

Reviewer #1

(Remarks to the Author)

I thank the authors for the changes to the manuscript. In my opinion, these changes improve clarity and make the story more distinct by including the antibiotic treatment in the main text/figures.

I would like to re-emphasize one minor point that I made previously. Some figures in the main text (and multiple in the supplement) do not include ticks on the y-axis. If this is the preferred style of the authors, this is ok with me as long as the grey line corresponds to the number on the y-axis. In some cases, it looks to me as if the line does not correspond to the number (e.g., Fig. 5 B, E) and in another panel, there is no line at all (e.g., Fig. 5C). The very least, I ask the authors to indicate ticks here and suggest the authors to systematically double-check their figures again for this. Otherwise, the data is not entirely clear to me as a reader.

Reviewer #2

(Remarks to the Author)

The authors have done a good job in responding to all of my original comments. The additional data in the SI and moving selected data from the SI to the main text have helped to make the narrative clearer. The changes to the introduction and the revised abstract have also greatly improved and clarified the rationale of the work.

I do like the new abstract, but I think it is important to state that HGT was observed to play a major role in the spread of resistance, and vertical inheritance dominated both in the presence and absence of selection. I suspect that people reading it will immediately wonder if HGT played a role.

I still think it would be useful on line 109 to state “with their respective *beta-lactamase encoding* antibiotic-resistant plasmids.” or similar, to make it clear that they provide resistance to ampicillin.

I have no additional comments.

Dear Editor,

Please find enclosed a revised version of the manuscript entitled "Multi-layered ecological interactions determine growth of clinical antibiotic-resistant strains within human microbiomes" (NCOMMS-24-74559). We would like to thank the reviewers for their thoughtful and constructive criticisms, which have helped us improve the clarity, robustness, and impact of the manuscript. We have carefully considered each comment and revised the manuscript accordingly to address all concerns raised during peer review. The changes are highlighted in yellow in the revised manuscript, and we provide a point-by-point response to each reviewer below.

Editor comments:

Thank you for your patience whilst our team made an editorial decision on your manuscript.

Thank you again for submitting your manuscript "Multi-layered ecological interactions determine growth of clinical antibiotic-resistant strains within human microbiomes" to Nature Communications. We have now received reports from 2 reviewers and, after careful consideration, we have decided to invite a major revision of the manuscript.

As you will see from the reports copied below, the reviewers find your work of interest and translational value. However, they raise important concerns that limit the strength of the study, and therefore we ask you to address them with additional work. In line with the reviewers comments, please provide further clarification and discussion, clearer rationale and further experimental work to strengthen your study and the findings reported within.

If you feel that you are able to comprehensively address the reviewers' concerns, please provide a point-by-point response to these comments along with your revision. Please show all changes in the manuscript text file with track changes or colour highlighting. If you are unable to address specific reviewer requests or find any points invalid, please explain why in the point-by-point response.

We would like to thank you for your time and for the opportunity to revise our manuscript. We are grateful for the constructive guidance provided by both reviewers and by your editorial comments. We have carefully considered all feedback and made substantial revisions to strengthen the manuscript. In particular, and in line with the points you raised, we have:

- Improved the clarity of the conceptual framework and justified the relevance of using clinical, resistant strains even when antibiotics are not present.
- Expanded and restructured the main figures, including moving previously supplementary data into the main text (e.g. antibiotic treatment results are now in Figure 3), to better support our conclusions across both antibiotic-treated and untreated conditions.

- Added new data and figures, including more detailed strain-level genomic comparisons (new Supplementary Figure 1), competitive growth assays with resident *E. coli* strains (Figure 4 and Supplementary Fig. 9), addressing multiple reviewer concerns.
- Clarified sample sizes, inoculum densities, community carrying capacity and the design rationale of the experiments, including new Supplementary Figure 8 to visualise initial conditions.
- Rephrased sections of the introduction, results and discussion to clarify the conceptual focus, better articulate the rationale for our approach, and acknowledge limitations where appropriate.

We believe these revisions have substantially improved the manuscript and hope it will now be suitable for publication in Nature Communications.

Reviewers comments:

Reviewer #1 (Remarks to the Author):

This manuscript by Leon-Sampedro et al., addresses the ability of clinically relevant, antibiotic-resistant *E. coli* strains to invade the intestinal microbiota of healthy individuals. The authors report on differences in growth of the *E. coli* strains within microcosms of healthy donors' intestinal microbiota. Further tests reveal associations between this phenotype with monoculture growth and competition with resident *E. coli* of the intestinal microbiota. What stands out to me is the very elegant experimental set-up and that is close to the clinic. Lots of work has been done and the individual experiments are sound. I do think that the authors are asking the right questions and I strongly believe that meaningful conclusions can be drawn from the work on this set up in the future. At this moment, I am afraid that I had difficulty in connecting some of the dots, as described in more detail below.

Major concerns

1. Conceptual framework. When starting to read this manuscript, I thought that one unique angle of this study would be the focus on antibiotic resistant *E. coli* isolates. Already after the first section of the results, it turns out that the statistically significant effect is observed in the absence of antibiotics and from this point onwards, it feels a bit as the antibiotics do not really matter anymore for the story and very similar observations could have been made by asking how a non-resistant *E. coli* strain, maybe even a lab strain, would colonize these microcosms. To be honest, I think such a simpler set up would already be challenging enough to disentangle this in a meaningful manner.

We thank the reviewer for this comment. We have now made four main changes to strengthen the rationale for our experimental set up and better integrate the results from the antibiotic-treated groups.

(1) We clarified in introduction and discussion the value of using clinical strains (L40, L78, L97, L402). Because these strains were isolated from hospital patients and carry clinically important resistance determinants, understanding their interactions with microbiota from humans is more relevant to the resistance problem than if we had used a lab strain.

(2) We explain in more detail the relevance of the antibiotic-free treatments (L40). Past research indicates the long-term spread or decline of resistance is strongly influenced by how resistant strains grow in the absence of antibiotics compared to other microbes, for example in asymptomatic carriers not undergoing treatment or in patients after treatment has finished. We added further explanation in the introduction (L40).

(3) We provide more detail on the motivation for using four strains rather than one (L97). This enabled us to detect both repeatability of some patterns across strains, and to identify differences among strains. This type of strain variability is relevant for the resistance problem because past work has shown certain types of resistant strains are particularly successful at colonising human microbiomes, and our data contribute to understanding that. We also now give more information about genetic differences among these strains (new supplementary figure 1).

(4) We agree with the reviewer that the results related to antibiotic treatment are also important. We have now moved these results into the main text (L236, Figure 3). These results support our overall conclusions and those from the antibiotic-free treatments. We hope this has strengthened the paper by now covering both scenarios (with/without antibiotics) more directly.

2. Distinction from other ongoing work. The point above becomes even more severe in light of reference 48. I appreciate that the authors bring up their other work. As I understand, the angle of using antibiotic-resistant *E. coli* is one point that distinguishes this from the other work. If the angle of antibiotics does not really matter much here, maybe the two stories are not that distinct any more or the story here would need to focus more on the antibiotics again to become distinct more clearly.

We thank the reviewer for this observation. We wanted to be transparent by mentioning that both works come from the same sampling collection and were conducted around the same time. However, the two studies address different questions. We have clarified this to bring out the novelty of the current manuscript more. First, we now place more emphasis on the antibiotic treatment groups, by bringing these data into the main text (L236, Figure 3), as described above. The other paper does not address antibiotics or resistance. Second, we clarified how the research questions addressed by the two papers differ (L525). Briefly, the other paper focuses on commensal *E. coli* strains isolated from healthy-human microbiomes, asking how these commensal strains are adapted to their original vs other microbiomes. In contrast, this study focuses on antibiotic-resistant strains isolated from hospital patients, asking how they interact with and change resident microbiota when they are newly introduced into a given microbiome. Thus, the focus on antibiotic resistance and invasion of human microbiomes by incoming resistant pathogens is distinct to this paper, and the main conclusions of the two studies are different.

3. Sample size and generalizable conclusions. I am wondering whether the current version of the manuscript tries to achieve too much with too small of a sample size. It is just really many variables and therefore I think hard to find meaningful associations or generalizable principles (different *E. coli* strains with different plasmids going into different healthy donors). Maybe one way out would be to keep the set up and sample size and focus more on one individual strain or donor or resistance plasmid and go more in depth and potentially try to test causation of one of the observed correlations or go for a more mechanistic understanding.

We thank the reviewer for raising this point. We would first clarify that the plasmids and strains are treated as a single unit in our study (the plasmids are present in their original bacterial hosts as isolated from hospital patients), reflecting what would happen in real clinical scenarios. We included multiple human donors simply so that we could identify repeatable patterns, rather than to investigate differences among individuals (that is more the focus of the other study mentioned above). Instead, our focus was on the differences between clinical strains and on understanding why some strains are more successful than others when facing a resident microbiota. This is also supported by our results: variability in invasion outcomes was mainly explained by differences between focal strains, not by differences between donors (first section of Results).

We appreciate the reviewer's suggestion about focusing on a single strain or donor to go deeper into mechanistic details. However, doing so would leave open the question of whether the observed results are specific to that particular strain or donor. We now discuss this aspect on L97. Nevertheless, we now include more details about genomic differences among strains (new Supplementary Figure 1), providing more context for the strain effects we identified.

Finally, we note that the scale of our experiment, including multiple clinical strains and multiple donor communities in a fully-factorial design, is already relatively high compared to many existing microcosm or simulated gut systems. We have now clarified this aspect including implications for future work on L97 and L483.

Minor comments

For multiple figures, please make sure outside ticks are added to all y-axes, it is not always clear what the numbers on the y-axis refer to. Especially in Fig. 2b, c, it is not entirely clear.

Done.

It might be worth rethinking which figures should go in the main text and which ones in the supplement, e.g. if the aim from fig. 2 onwards is to find associations/correlations with the phenotypes observed in the first figure, wouldn't it help to include the data on these associations in the respective subsequent figures (e.g. move Fig. S2 to the main Fig. 2) rather than having it in the supplement?

We agree that in places the scatterplots make it easier to see relationships between key variables. We moved the figures in some cases to account for this (Fig. 2 and Fig. 4). We note that, because such associations arise from the variability among different strain-donor combinations (noted for example on L170), it is important to place them in the context of strain- and donor- effects, which are shown more clearly by the accompanying panels in the relevant figures. We hope the new figures therefore bring out more clearly both the differences among strains/donors and correspondence between output variables from different experiments.

L. 183, use of the term "interaction", I am aware that this could become a bit philosophical. Is it truly an "interaction" to see the addition of one strain go along with the composition of the community composition in the slurry?

We agree the term "interaction" requires more careful explanation and can sometimes imply direct mechanistic processes, which we do not directly observe in this study. Our intention was to use the term in a broader ecological sense, referring simply to one strain or species affecting the growth or abundance of others, as used in some relevant past work in microbial ecology. In this case, this is reflected by the introduction of a focal strain leading to changes in community composition. We added citations and clarified the text (L201).

I. 73, for clarification, it might arise elsewhere, where did the four clinical strains come from? E.g. from patients in the same area than the healthy individuals? Is there a real chance of the healthy individuals in this study acquiring these resistant strains?

Yes, the strains were sampled from patients in Basel, Switzerland, approximately 90km from Zürich, where the healthy donors were sampled. We added more details to the text (L508).

Fig. 1

- panel A, the label of the four strains is not entirely clear to me, does the end of the black line indicate where they are in the tree or rather which phylogroup they belong to. I am also not clear about the heading "clinical resistant strains", are all strain in the tree clinically relevant or only the four in the focus on this study?

Fixed. We have clarified now the Figure 1.

- panel B, is it really n=90 microcosms? I do not fully understand the number or the base on which it was calculated.

Yes, we added the detailed numbers in the figure legend: Total number of microcosms (n=90; 4 focal strains + 1 uninvaded condition × 3 samples from healthy donors × with antibiotic/ untreated condition × 3 replicates).

- What was the rationale for this exact concentration of 72.ug/ml? It might be worth to discuss the implications of the antibiotic on the microbiota per se more in the later sections.

We used 7.2 µg/ml because this concentration does not fully inhibit susceptible *E. coli*, allowing partial survival of the resident members of the community and thus enabling interactions with resistant strains. This is consistent with previous work in this system (now cited on L566). We agree that changing the concentration could affect the outcomes, for example, stronger selection at higher concentrations may amplify the observed effects, which we now discuss on L464.

- I. 104 "In the presence of ampicillin, all four introduced resistant strains underwent net positive population growth over 48h (Fig. 1C)." Is this backed up by statistics?

Yes. In the presence of ampicillin, all four introduced resistant strains underwent net positive population growth over 48h (Fig. 1C) (paired t-test; Ec040: $t = 15.1$, $df = 2$, $p = 3.6 \times 10^{-7}$; Ec069: $t = 8.82$, $df = 2$, $p = 2.2 \times 10^{-5}$; Ec131: $t = 3.65$, $df = 2$, $p = 0.0065$; Ec744: $t = 5.35$, $df = 2$, $p = 6.8 \times 10^{-4}$). We added details to the text on L114.

Fig. 2

- Fig. 2b, what is the number of replicates and what does the horizontal line show? The figure legend says median, to me it looks like the mean.

Done.

- Fig. 2b/c, what is the number of replicates here? $n=4$? Would be good to add this to the figure legend.

Done.

Fig. 4, - is there data on the competing strain? Might be good to write out based on which exact data or kind of data the term "competition" is being used.

Thank you for your comment. The term "competition" refers to pairwise competition experiments that we performed in co-cultures between the focal resistant strains and resident *E. coli* strains isolated from the three communities. We have now clarified this on L288. We also added the requested data on the competing strains by incorporating additional details into the main manuscript (L288) and adding a new Supplementary Fig. 9a, showing growth of both focal strains and competitors during these experiments.

- what was the rationale of doing 5 replicates here (and 3 replicates for figure 1)? I do think that the analyses of these experiments will be very sensitive to the number of replicates. If 5 replicates would be done in Fig. 1, I could imagine a modified outcome of the statistics done there. In general, I think the more replicates the better.

The main experiment (Figure 1) was limited by the scale and logistical complexity of the setup. We worked with 90 Hungate tubes under strict anaerobic conditions. Each sample required careful handling, time-sensitive processing, and anaerobic transfers. This constrained the total number of replicates we could feasibly manage while maintaining experimental quality and consistency. In contrast, the experiment in Figure 2 involved a smaller and technically simpler setup, allowing 5 replicates per condition.

We agree increasing the number of replicates would strengthen statistical power. Nevertheless, the consistency of key effects across experiments (for example, the strain effect observed in Fig. 1 and Fig. 2) is unlikely to have arisen due to low numbers of replicates (because low sample sizes would be expected to increase noise in the data and make such correspondence across experiments less likely). We also note that by using a fully-factorial design (every combination of strain and donor tested with multiple replicates), each strain is in fact tested in 9 microcosms (3 per donor) and each donor sample in 12 microcosms (3 per strain) in both the absence and presence of antibiotics. This means that testing the main effects (for example, differences among strains, one of our key results) is based on a comparison across more than only three replicates per strain, because it incorporates the data from all donors. Finally, despite having only three replicates per combination, our main results, such as the poor performance of Ec744 in Fig. 1, were consistent across replicates within treatment groups.

Fig. 5

- l. 288, what was the authors' rationale to look into carbon sources? was it based on existing literature or based on existing data on their set up that carbon sources would be limited. Wouldn't we only expect to see a correlation here if carbon sources would be of limited supply in the experimental set up?

Our rationale for testing metabolic profiles using BioLog plates was based on existing evidence that metabolic overlap between incoming strains and resident microbiota can limit colonization success. Strains with broader metabolic capabilities, or with unique nutrient-use profiles, might experience reduced competition and therefore grow more successfully across microbiomes. In particular, Spragge et al. (Science 2023) showed that nutrient competition, where resident communities deplete resources needed by pathogens ("nutrient blocking"), is an important barrier to colonization. We have added further details in the text (L329).

I am making this point here, I think it applies also a bit more generally that the associations that are being tested for do come a bit out of the blue and seem driven a lot by practical reasons (e.g. available biology system) rather than careful consideration of what competing resources the individual E. coli strains could face in the very specific system that is under study.

We used BioLog plates primarily as a practical and standardized method to characterize metabolic traits across strains. While not designed to replicate the exact nutrient environment of our microcosms, the BioLog test allowed us to assess broad variation in carbon-source utilization, testing for metabolic versatility, which is known to influence competition and colonization dynamics in microbial communities. We have now added more references and revised the discussion (L445) to incorporate this and the reviewer's point above about the role of this type of competition depending on what is limiting. In this context, we believe our analysis provides useful insight into the relevance and limitations of metabolic data when understanding ecological outcomes in complex systems (noted on L445).

- Fig. S13/l. 329 ff I do agree that this is interesting, it seems a bit disconnected though. E.g. the inhibition phenotype was observed on plates and the microcosms experiments were done in liquid. Based on this difference alone, there could be many reasons why there is no association.

We agree with the reviewer's comment (L310). We have now moved the agar-inhibition-assay data to the supplement (Supplementary Fig. 10.). We did not remove it completely, because it provides new information about inhibitory interactions affecting clinical resistant strains, but we agree it is not fully connected to the rest of the story and therefore works better as supplementary material.

- Fig. S14/S15, l. 340 ff I see the point and the work and I think crucial to get back to the original research question again. In my view, it somehow does not help the story much in its current form and rather seems like an add on, maybe because somebody asked for this, rather than a possibility that was taken into account from the beginning or was in mind from the start. In general, I like the set-up of using a lab strain with the different plasmids because it goes in the direction of disentangling which of the many variables (plasmid, strain, host community) in the original set up do play a role and make a difference. As I wrote at the beginning, I think there is tremendous potential in the experimental set up and as exciting as it might be to address all these questions that are raised directly or more indirectly in this manuscript, it might be easier for this reader if there would be a clear story line that addresses one or fewer questions more consistently.

We thank the reviewer for recognising the work and the strength of our experimental setup. To clarify, screening for plasmid transfer was part of our original experimental setup. In the main microcosm

experiment, we systematically tested 1440 colonies with relevant resistance phenotypes to detect possible transconjugants. All tested colonies matched the incoming resistant strains, indicating no detectable plasmid transfer under those conditions. To explore whether transfer was possible but under the detectable limit, we complemented the main experiment with a follow-up assay using a counter-selectable *E. coli* donor. This showed that two of the plasmids could transfer into resident strains in the same gut microcosms, although at very low frequency (Supplementary Fig. 15). We have now edited the Results and Discussion (L374 and L460) to better integrate these findings into the main story and to introduce the role of horizontal transfer earlier in the text (L36).

Reviewer #2 (Remarks to the Author):

In this manuscript, the authors track the ability of four clinically isolated *Escherichia coli* (Ec) harbouring antibiotic resistance plasmids to invade human faecal microbial communities in vitro. The authors hypothesise that different strains will have different abilities to invade resident communities, likely due to differing intrinsic growth abilities, interspecific interactions or niche overlap.

The authors introduce the focal Ec strains into communities with and without positive selection for the focal MDR Ec strains (ampicillin treatment). In the presence of positive selection, all of the focal strains are able to maintain positive growth and invade the community, however in its absence the results are variable, with some strains unable to become established. The leading result is that the intrinsic growth ability of the strains (i.e. their growth rate in the absence of the communities) is highly correlated with their ability to invade the community. In contrast, niche overlap and interference competition with other resident Ecs are poor predictors of the growth ability of the focal strains in the presence of the communities.

While the presented work does advance our understanding of the criteria that enable resistant pathogenic bacteria to become established within complex multispecies communities, I have some concerns that require addressing in the current manuscript.

Major comments:

1. It is unclear to me why are the inoculum densities so different between the sterile (figure 2a) and live (figure 1c/d) microcosms. Are the focal strains unable to invade if inoculated at low density? Can these experiments be directly compared given the $\sim 10,000$ x difference in inoculum density ($\sim 10^3$ vs $\sim 10^7$)?

We thank the reviewer for pointing this out. First, to clarify: in the live microcosms (Fig. 1c/d), the inoculum size was $\sim 10^5$ CFU/ml, not $\sim 10^7$. The densities approaching 10^7 shown in the plot correspond to the first sampling point, 2 hours after inoculation. We now realise this needed clearer explanation and have updated the figure accordingly by adding a break in the x-axis to indicate the 2-hour gap between inoculation and first sampling. The timing of the first sampling (and, where applicable, antibiotic addition) at 2h was chosen for consistency with past work in this system. We have clarified this point in the revised manuscript (L149).

The rationale for using a lower inoculum size in the sterile microcosm experiment (Figure 2a) was to allow a longer growth phase of the introduced strains, which we expected to increase the sensitivity of the assay to detect differences between strains that could otherwise be masked at higher starting densities. We added text to clarify this on L603. A further difference between the two experiments is that the experiment in Fig. 1 included a passage to fresh microcosms after 24h. Thus, the two experiments are not intended to be directly compared in terms of inoculum density, but rather serve complementary purposes: the live microcosms test invasion success in the presence of a complex community, while the sterile microcosms test the ability of the strains to grow in the absence of community-mediated effects.

2. Given the very high inoculum density in the live microcosms who is invading who? It is unclear what the initial density of the faecal community is at T₀ (final densities are reported in S9). But presumably this very high inoculum density is much higher than any one individual strain/species within the mixed faecal communities.

We agree that the initial ratio of focal strain to resident community is essential for interpreting invasion dynamics, and we have now clarified these starting conditions in the revised Supplementary Figure S8. Each focal strain was introduced at an inoculum of approximately 1×10^5 CFU/ml, which is substantially lower than the starting abundance of the resident microbial communities after dilution. Based on direct measurements at 2 hours (prior to significant growth), total community abundance ranged from approximately 6×10^8 to 1.6×10^9 CFU/ml across the three donor-derived communities. Focal strain abundances after 2 hours ranged between 1×10^5 and 8×10^6 CFU/ml. We now present these data in Supplementary Figure S8 (panels E and F), where focal strain abundances are shown alongside total community abundances, and a shaded band marks the abundance range corresponding to taxa that comprise 0.005% to 14% of the community. For context, the most abundant taxon in these communities typically represented ~14% of the total population, equivalent to $\sim 2.2 \times 10^8$ CFU/ml. These additions clarify that the introduced focal strains typically constituted less than 1% of the total microbial abundance shortly after inoculation and were consistently less abundant than the dominant resident taxa.

3. In the absence of antibiotic selection is there space for the focal Ec strains to expand into or are the populations already at carrying capacity at T₀? This may explain why there is net positive growth in the addition of antibiotics. Ab treatment is likely killing a substantial part of the resident population allowing the focal Ec to expand into, whereas in the absence net growth is close to zero (or below) for strain/community combinations. This may help to explain why Ec744 is unable to be maintained in the absence of selection, given its relatively low

Regarding carrying capacity at T₀: the initial total bacterial abundance of 10^7 cells/mL increased after 2 hours to approximately 10^9 cells/mL (see new Supplementary Fig 8), of which *E. coli* made up only a fraction (Supplementary Fig 8). This suggest there was net positive growth at the level of the entire community, as well as the introduced focal strains, and this was observed both with and without antibiotic treatment and with and without the introduction of resistant focal strains. Nevertheless, antibiotic treatment did lead to changes in community composition (shown in Supplementary Figure 6.).

We agree that antibiotic treatment could, in principle, create ecological space by inhibiting competing resident microbiota and thereby facilitate the spread of resistant focal strains. Despite this, our past work showed there can still be strong suppression of incoming resistant bacteria by resident microbiota in the presence of antibiotics (Letten et al., ISME J 2021; cited on L407). Our data are consistent with this in showing that antibiotics affect community composition but do not suppress all taxa evenly. We revised the discussion to clarify that this type of competitive release may contribute to the positive growth of Ec744 with vs without antibiotics (L404).

4. The authors state that interactions can be deduced from differential changes in the relative abundance of members of the resident community. While I agree that this is true for some interactions, others such as commensalism (+/0) and amensalism (-/0) would not result in changes in the relative abundance of community members. Therefore, you cannot rule out interactions playing an important effect simply because the community composition does not change. This may explain why significant changes in community structure were not observed in Donor1 despite it having an inhibitory effect on three out of the four focal species.

We agree that not all ecological interactions necessarily lead to detectable changes in relative abundances. We have now acknowledged this limitation explicitly (L204). We also note that our approach incorporated both comparisons among treatment groups inoculated with different resistant strains (L210) and testing for differences relative to uninvaded control microcosms (L220). With the latter, we detected which community members increased or decreased upon introduction of a focal strain, including with Donor 1's sample. While we recognize that other forms of interactions may occur without detectable compositional changes, the relative abundance shifts we observed do provide meaningful ecological information. For example, after the invasion of Ec040 into Donor1 microcosms, we observed a marked suppression of Lachnospiraceae (L234), but this was not observed with other strains, consistent with their relatively poor invasion success in this sample.

Minor comments:

1. Line 28-29: "success was positively associated with... their competitive interactions with resident E. coli." The data does not convincingly show this. In fact, the authors state on line 333, "the patterns of inhibition were not closely associated with growth success in the live microcosm system".

We would like to clarify that the two parts of the manuscript the reviewer refers to are based on different experimental setups and different types of data. The statement on lines 28–29 refers to the correspondence between results from pairwise competition experiments in liquid cultures and invasion success in the live microcosms. In contrast, the statement on line 333 refers to separate experiments testing for inhibition in agar overlay assays. The agar overlay assays are less connected to the other experimental conditions we tested, so we have now moved those data to the supplement (see our response to a similar point by the other reviewer above). We also clarified how competitive ability is inferred in the other experiments (L289 in the results).

2. Presumably ampicillin resistance is provided by the resistance plasmids that the focal Ec's carry, this is not made completely clear at the start of the results.

We agree that this point was not clearly at the beginning of the Results section, and we have now clarified that the ampicillin resistance observed in the focal *E. coli* strains is conferred by plasmids they carry (L574). As detailed in Supplementary Table 3, each focal strain harbours a distinct plasmid encoding specific β -lactamase genes—including bla_{OXA-48}, ESBL-type, or other carbapenemase genes that confer resistance to ampicillin and related antibiotics. To track plasmid stability and strain abundance under different conditions, we used a combination of non-selective chromogenic LB agar and three chromogenic selective media (OXA, CARBA, and ESBL; bioMérieux). These media are designed to detect phenotypic resistance to particular classes of β -lactams by combining antibiotic selection with chromogenic substrates. In our system each focal strain's resistance phenotype corresponds uniquely to the presence of a single plasmid carrying a known β -lactamase gene (see Supplementary Table 3). Loss of plasmid carriage results in loss of resistance, and therefore no growth on the corresponding medium, allowing plasmid stability to be inferred from growth/no-growth patterns on these plates.

3. The phrase “more differently” on line 115 is vague; consider revising.

We have rephrased this (now L125).

4. Figure S1 – the use of box plots with three data points in this figure seems redundant as the data points just fall on the whiskers and median line.

Solved.

5. Figure S2 – I suspect you would see a very similar positive correlation between the final abundances of the sterile cultures and the antibiotic-treated cultures because the carrying capacity of Ec744 strain is lower than the other strains.

The reviewer is correct, there is indeed a positive correlation between final abundances in sterile cultures and antibiotic-treated cultures (see plot below). This is consistent with our other observations for Ec744. We note that our data also cover other parts of the growth cycle, not only carrying capacity. As shown in Supplementary Figure 3, differences among strains are also apparent at earlier stages of the growth cycle, and in fact in some microcosms they do not fully plateau after 24h. Therefore, while the positive correlation suggested by the reviewer exists, it supports our interpretation that intrinsic growth ability, not only carrying capacity, plays a key role in explaining differences in strain success across conditions.

6. Line 90-92: I appreciate that the authors use clinically isolated strains but they justify their use by stating, “the use of clinical strains and plasmids accounts for possible genomic factors that may influence colonisation success, such as pathogenicity islands, prophages or other plasmids, which are often not represented in domesticated or model strains.” But how does the use of four strains account for this? The role of genomic differences was not examined, and there likely isn’t enough data to conclude how such specific genomic differences between the strains alter the outcomes presented here.

We agree that with only four strains, it is not possible to fully disentangle the effects of specific genomic features on colonization success. Our intention was not to systematically study the role of individual genomic elements, but rather to account for realistic genomic variability by using clinical isolates that carry diverse accessory genomes, including different plasmids, prophages, and potentially pathogenicity islands, which are often absent from lab strains. To maximize this genomic diversity, we deliberately selected strains that were phylogenetically distinct, as shown in Figure 1. We now provide more details on the genomic content and differences among the strains (new Supplementary Figure 1). The rationale was to identify patterns repeatable across phylogenetically distinct clinical strains with realistic genomic backgrounds (L97). Thus, while we cannot directly attribute specific outcomes to individual genomic features, the use of diverse clinical strains allowed us to capture a more realistic range of strain behaviors relevant to clinical contexts.

Reviewer comments

Reviewer #1

“Some figures in the main text (and multiple in the supplement) do not include ticks on the y-axis... In some cases, it looks to me as if the line does not correspond to the number (e.g., Fig. 5 B, E) and in another panel, there is no line at all (e.g., Fig. 5C).”

We thank the reviewer for pointing this out. We have now carefully revised all figures in the main text and Supplementary Information to ensure that y-axis ticks are included or adjusted where necessary. In the specific cases of Fig. 5B, 5C, and 5E, we have corrected the grey lines and ensured they align accurately with the corresponding y-axis values.

Reviewer #2

“I do like the new abstract, but I think it is important to state that HGT was observed to play a major role in the spread of resistance, and vertical inheritance dominated both in the presence and absence of selection.”

We have now revised the abstract to include a brief but clear mention of horizontal gene transfer. Specifically, we added the sentence:

“We also detect horizontal transfer of resistance plasmids in some conditions, but transconjugants remain rare across treatments.”

“I still think it would be useful on line 109 to state ‘with their respective beta-lactamase encoding antibiotic-resistant plasmids.’ or similar...”

We have now modified the sentence to read:

“...with their respective beta-lactamase-encoding antibiotic-resistant plasmids.”

We thank the reviewer for this suggestion.